# A multiscale analysis of mean-field transformers in the moderate interaction regime

**Giuseppe Bruno**
Department of Mathematics and Statistics
University of Bern
giuseppe.bruno@unibe.ch

**Federico Pasqualotto**
Department of Mathematics
University of California, San Diego
fpasqualotto@ucsd.edu

**Andrea Agazzi**
Department of Mathematics and Statistics
University of Bern
andrea.agazzi@unibe.ch

## Abstract

We study the evolution of tokens through the depth of encoder-only transformer models at inference time by modeling them as a mean-field interacting particle system, and analyzing the corresponding dynamics. More specifically, we consider this problem in the *moderate interaction regime*, where the number $N$ of tokens is large and the inverse temperature parameter $\beta$ of the model scales together with $N$. In this regime, the dynamics of the system displays a multiscale behavior: a fast phase, where the token empirical measure collapses on a low-dimensional subspace, an intermediate phase, where the measure further collapses into clusters, and a slow phase, where such clusters sequentially merge into a single one. We characterize the limiting dynamics in each phase, exemplifying our results with some simulations.

## 1 Introduction

The transformer architecture [49], through its extensive use in Large Language Models, has played a crucial role in the recent, unprecedented developments in machine learning and artificial intelligence. One of the key innovations at the heart of this architecture are self-attention modules [6], allowing to capture long-range dependencies in the data, e.g., in prompts with a large number $N$ of tokens. To further improve their performance, practitioners have implemented these models in different hyperparameter regimes, e.g., choosing the model's inverse temperature parameter $\beta$ (a parameter that scales the query-key dot products in the self-attention layer) as a function of $N$ [37, 43]. However, the groundbreaking empirical success of these machine learning models remains largely unexplained from the theoretical perspective. In particular, a precise mathematical description of the internal representations learned by transformers, and of how these representations behave in different hyperparameter regimes, is still lacking.

A promising approach to fill this gap was presented in the work [46], where the authors interpret tokens traveling through a deep stack of transformer layers as particles evolving in time and interacting in a mean-field way. A subsequent line of work [24] has then observed that tokens in this model tend to organize into clusters, offering - in a simplified setting - a compelling qualitative explanation of how transformer models build representation of complex input data.

Despite its apparent simplicity, this interacting particle system exhibits a remarkably rich dynamical behavior. Indeed, recent studies have identified distinct dynamical phases, characterized by quali-

tatively different clustering patterns, which depend on specific choices of parameters, timescales, and initial conditions. However, these results often rely on restrictive - and sometimes unrealistic - assumptions, and they typically capture only limited aspects of the collapse dynamics, providing partial views of the transformer's complex dynamical landscape that are difficult to reconcile into a consistent and global dynamical picture. Such a global characterization of the clustering phenomenon in a realistic parameter regime is arguably fundamental to understanding how internal representations form in deep models and how to operate such models in the optimal hyperparameter regimes.

**Contributions**   In this work, we study the dynamics of the mean-field transformer model developed in [46, 25], constrained on the $d$-dimensional sphere, in the limit of large context size, i.e., when the number $N$ of input tokens is large. Motivated by recent scaling strategies in state-of-the-art LLMs such as SSMax [37] and YaRN [43] (used respectively in Llama 4 and Qwen 3), we consider the setting where the inverse temperature parameter $\beta$ grows with $N$. In this regime, our contributions can be summarized as follows:

1. We identify three distinct dynamical phases (respectively denoted the *alignment*, *heat* and *pairing* phases), corresponding to different scales of time as a function of the parameter $\beta$. In each phase the model dynamics displays, asymptotically in $\beta$, qualitatively distinct behavior, characterized by a different limiting equation.

2. In the alignment phase, occurring on a fast timescale of order $O(1)$, we prove our main technical result: under general assumptions on the parameter matrices, the finite particle dynamics converges to a linear transport equation modeling the collapse of the token measure onto a low-dimensional manifold dictated by the spectral properties of such matrices. To the best of the authors' knowledge, this phase was not yet identified in the literature.

3. In the heat and pairing phases, occurring respectively on timescales of order $O(\beta)$ and $O(e^{c\beta})$, we identify, under stronger conditions on the parameter values, the limiting dynamics as a forward or backward heat equation on the aligned manifold (leading, in the backward case, to metastable clustering) and a finite-dimensional system of ODEs describing sequential cluster merging along geodesics.

Together, these phases reconcile various previously identified dynamical regimes as different timescales of a single unified dynamical picture. Furthermore, our multi-phase analysis allows to relax some of the restrictive assumptions imposed by previous works, extending their applicability to more realistic scenarios.

**Related works**   The model studied in this paper was introduced in [46], where the authors also identify the heat equation as describing the dynamics of the particle system in $\mathbb{R}^d$ in the large $\beta$ regime. This limit emerges as a correction term in their analysis upon subtracting an appropriate leading order term from the prelimit equation. In this paper, by considering the dynamics on the sphere – resulting from the inclusion of the layer normalization in our model – we provide a justification for the spontaneous collapse of the system's state to a subspace where this correction term becomes of leading order, dominating the dynamics of the model on a certain timescale.

In [25, 24, 29, 13, 44], the authors identified the clustering behavior occurring in this and closely related models as $t \to \infty$ for $\beta, N$ fixed. Analogous convergence results, under different assumptions, are provided in [18, 34], while quantitative contraction rates for such convergence are given in [16]. These works, however, do not address the dynamically meta-stable phases characterized by partial clustering numerically highlighted in [25]. This intermediate phase is explored in [9] in the large $N$ limit for tokens distributed uniformly at initialization and in [23], where the authors study the formation of meta-stable clusters under the assumption that the system is initialized into well-separated configurations. Furthermore, in [23] the authors identify different dynamical timescales in the finite $N$ case and characterize for the first time what we refer to in this paper as the pairing phase in the large $\beta$ limit. In all these cases, however, the results are limited to the setting where the model's key, query and value matrices, $Q, K$ and $V$, were multiples of the identity. More recently, the work [11] analyzes the stability of fixed points of the same model based on the eigendecomposition of $Q, K, V$ under the weaker assumption that parameters satisfy a modified Wasserstein gradient flow condition ($Q^T K = V = D$), but does not study the dynamical landscape connecting such fixed points. Finally, in [3, 4], the authors discuss clustering for hardmax transformers. Our work provides a framework to combine the observations listed above in a unique dynamical picture.

Our modeling approach shares conceptual roots with the neural ODEs literature [14, 21]. However, a key distinction is that we consider $N$ particles interacting through a mean-field PDE, as opposed to one in the previous references. This connects our work to the broader literature on mean-field models for neural networks [45, 36, 17, 2, 19], where timescale analysis has also been a subject of interest (see [7]). In contrast to these works, which typically focus on training dynamics, our study centers on the inference-time evolution of representations through network depth.

Finally, our research relates to the study of moderate scaling limits in interacting particle systems. For instance, Oelschläger [39] proved the convergence of certain systems to the porous medium equation with noise. These results were subsequently extended to cases without noise [41, 40], with different exponents [22], or employing different techniques and equations [12, 10, 42]. Another relevant line of work investigates the convergence of specific interacting particle systems to the heat equation, explored both numerically and theoretically [20, 8, 33].

## 2 Framework and notation

We consider the framework introduced in [46, 25, 24], modeling the transformer architecture as a discrete-time dynamical system governing the evolution of $N$ *tokens* $\{x_i(\ell)\}_{i=1,\dots,N}$ through its layers via:

$$\begin{cases} x_i(\ell+1) = \mathcal{N}\left(x_i(\ell) + \frac{1}{Z_{\beta,i}(\ell)} \sum_{j=1}^N e^{\beta\langle Q_\ell x_i(\ell), K_\ell x_j(\ell)\rangle} V_\ell x_j(\ell)\right), \quad \ell = 0,\dots,L-1, \\ x_i(0) = x_i, \end{cases} \tag{1}$$

where $\mathcal{N} : \mathbb{R}^d \to \mathbb{S}^{d-1}$ denotes the normalization operator on the $d-$dimensional unit sphere $\mathbb{S}^{d-1}$, $L$ denotes the depth of the transformer architecture and $Z_{\beta,i}(\ell) = \sum_{j=1}^N e^{\beta\langle Q_\ell x_i(\ell), K_\ell x_j(\ell)\rangle}$ is a normalization constant. The dynamics depends on the parameters with matrix values $Q_\ell$, $K_\ell$, and $V_\ell$ that represent the query, key, and value matrices at each layer, respectively.

In the spirit of neural ODEs [14], the authors then consider the infinite-depth limit of (1), leading to the following continuous-time model, describing the evolution of $x_i(t) : [0,\infty) \to \mathbb{S}^{d-1}$:

$$\dot{x}_i(t) = P_{x_i(t)}\left(\frac{1}{Z_{\beta,i}(t)} \sum_{j=1}^N e^{\beta\langle Q_t x_i(t), K_t x_j(t)\rangle} V_t x_j(t)\right). \tag{SA}$$

Here and throughout, $P_x y := y - \langle x, y\rangle x$ denotes the orthogonal projection of $y$ onto the tangent space $T_x \mathbb{S}^{d-1}$, $\langle \cdot, \cdot \rangle$ is the Euclidean inner product in $\mathbb{R}^d$, and $Z_{\beta,i}(t) = \sum_{j=1}^N e^{\beta\langle Q_t x_i(t), K_t x_j(t)\rangle}$ is the time-dependent normalization factor. The parameter $\beta > 0$ is interpreted as the *inverse temperature*.

**Remark 2.1.** *The MLP would act as a drift term in the dynamics, whose consequence should still be investigated further. We expect different dynamical behavior depending on the relative scale of the MLP coefficients and the attention part. Although the framework allows for the inclusion of feedforward layers via a Lie-Trotter splitting scheme (see [26]), we choose to isolate exclusively the self-attention mechanism, both because our interest lies specifically in its dynamics, and for the sake of clarity. For the same reason, we assume, as in the works cited above, the parameter matrices to be shared across layers: $Q_t \equiv Q$, $K_t \equiv K$, and $V_t \equiv V$.*

As the positional information of each token is encoded in its initial condition, the dynamics (SA) is invariant under permutations of the particles' indices. This symmetry allows us to fully characterize the system's state through the particles' empirical measure $\mu(t) := \frac{1}{N}\sum_{i=1}^N \delta_{x_i(t)}$, where $\delta_x$ denotes the Dirac measure centered at $x$. The measure $\mu(t)$ evolves according to the continuity equation:

$$\begin{cases} \partial_t \mu + \mathrm{div}(\chi_\beta[\mu]\,\mu) = 0 & \text{on } \mathbb{R}_{\geq 0} \times \mathbb{S}^{d-1}, \\ \mu|_{t=0} = \mu(0) & \text{on } \mathbb{S}^{d-1}, \end{cases} \tag{2}$$

where the vector field $\chi_\beta[\mu] : \mathbb{S}^{d-1} \to T\mathbb{S}^{d-1}$ is defined as

$$\chi_\beta[\mu] = P_x\left(\frac{1}{Z_{\beta,\mu}(x)} \int_{\mathbb{S}^{d-1}} e^{\beta\langle Qx, Ky\rangle} Vy\,\mathrm{d}\mu(y)\right), \tag{3}$$

with $Z_{\beta,\mu}(x) := \int_{\mathbb{S}^{d-1}} e^{\beta\langle Qx, Ky\rangle}\mathrm{d}\mu(y)$. This formulation extends the token dynamics to a flow on the space $\mathcal{P}(\mathbb{S}^{d-1})$ of probability measures over the sphere $\mathbb{S}^{d-1}$, encompassing both empirical and absolutely continuous distributions.

## 3   Main results

As discussed in the introduction, in this paper we consider the limit as $\beta \to \infty$ of the dynamics (3). To present the dynamical scales arising in this limit, consider a formal Taylor expansion of the vector field $\chi^\beta[\mu]$ generated by a sufficiently smooth measure $\mu$:

$$\chi_\beta[\mu](x) \approx \underbrace{\frac{\int e^{\beta\langle x',y\rangle} P_x V y \,\mu(x')\sigma(dy)}{\mu(x') \int e^{\beta\langle x',y\rangle}\sigma(dy)}}_{\text{(I)}} + \underbrace{\frac{\int e^{\beta\langle x',y\rangle} P_x V y \langle y - x', \nabla\mu(x')\rangle \,\sigma(dy)}{\mu(x') \int e^{\beta\langle x',y\rangle}\sigma(dy)}}_{\text{(II)}}, \qquad (4)$$

where $\sigma$ denotes the Lebesgue measure on $\mathbb{S}^{d-1}$ and $x' = K^\top Q x$. For large $\beta$, Laplace approximation suggests that (I) typically dominates (II) at initialization, giving rise to a first, fast dynamical phase:

- **Alignment Phase**: on a timescale of $O(1)$, the limiting dynamics are governed by a linear transport equation (Eq. (5) below) and the token distribution rapidly collapses onto a lower-dimensional subspace determined by the spectral properties of the matrix $VK^TQ$.

After the dynamics collapses to this low-dimensional subspace, we identify some classes of parameters for which the leading-order contribution to the vector field, approximated by term (I), vanishes. In such scenarios, the dynamics becomes governed by term (II), which involves the gradient of the measure $\mu$. This gives rise to a second, intermediate phase:

- **Heat Phase**: operating on a timescale of $O(\beta)$ (achieved by rescaling time as $t' = t/\beta$), the dynamics within the previously identified subspace exhibits diffusive or anti-diffusive behavior. Depending on the model parameters (specifically the sign related to $VK^TQ$ restricted to the subspace), this phase can lead to further concentration into distinct clusters (backward heat equation) or to smoothing/spreading of the distribution (forward heat equation).

In the attractive case, we identify the limiting dynamics up until the formation of clusters. We expect the clusters to be invariant in this timescale, and to interact only on much longer ones.

- **Pairing Phase**: on an exponentially long timescale in $\beta$ (e.g., $O(e^{c\beta})$ for some $c > 0$, where $c$ depends on the distance between clusters), the clusters formed in the previous phase sequentially merge. Typically, the closest pair of clusters collapses first, governed by a system of ODEs describing their interaction, eventually leading to a single clustered state.

We refer to Appendix E for a graphical representation of the three phases introduced above.

We outline the structure of the remainder of this Section. In in Section 3.1, we recall a quantitative result connecting the large $N$ behavior of the ODE system with the behavior of the corresponding PDE in the relevant timescale. This will allow us to focus solely on the PDE analysis when we describe the three main dynamical phases in Sections 3.2, 3.3, and 3.4.

### 3.1   Large $N$ convergence

To connect the timescales analysis above to the $N$-particle system of ODEs (SA), we consider the regime where $N \to \infty$ and $\beta = \beta_N \to \infty$ *slowly enough with respect to $N$*. This is relevant for context scaling techniques and LLMs (see introduction). To proceed, we use the following lemma:

**Lemma 3.1.** *Assume that the initial tokens $\{x_i(0)\}_{i\in[N]}$ are sampled independently and identically distributed from a reference measure $\mu_0 \in \mathcal{P}(\mathbb{S}^{d-1})$. Let $\mu_t^{N,\beta}$ be the empirical measure for particles $\{x_i(t)\}_{i\in[N]}$ evolving via the ODEs (SA), and let $\mu_t^\beta$ be the solution to the continuity equation (2) with initial condition $\mu_0$. Fix a time interval $[0, T_\beta]$ where $T_\beta$ is a $\beta$-dependent timescale. If $\beta = \beta_N$ depends on $N$ and diverges slowly enough as $N \to \infty$, then:*

$$W_1(\mu_t^{N,\beta}, \mu_t^\beta) \to 0 \quad as \ N \to \infty,$$

*uniformly on $[0, T_\beta]$.*

*Proof.* This follows from the Dobrushin-type stability estimate: $W_1(\mu_t^{N,\beta}, \mu_t^\beta) \leq W_1(\mu_0^N, \mu_0)\, e^{L_\beta t}$, where $L_\beta$ is a positive constant depending on the Lipschitz constant of the vector field $\chi_\beta$, as discussed in [9]. The claimed convergence follows from $W_1(\mu_0^N, \mu_0) \to 0$ provided that $L_{\beta_N} T_{\beta_N}$ grows sufficiently slowly with $N$ such that the overall term tends to zero. $\qquad\square$

Our goal is to understand the behavior of $\mu_t^{N,\beta}$ in the joint limit $N, \beta_N \to \infty$. We denote the limiting distribution of $\mu_t^\beta$ as $\beta \to \infty$ by $\mu_t^\infty$. Following an argument analogous to that in [22], though in a different setting, we can decompose the convergence problem as:

$$W_1(\mu_t^{N,\beta}, \mu_t^\infty) \le W_1(\mu_t^{N,\beta}, \mu_t^\beta) + W_1(\mu_t^\beta, \mu_t^\infty).$$

In our regime, Lemma 3.1 guarantees that the first term vanishes as $N \to \infty$. Consequently, the analysis of the $N$-particle system in this coupled limit reduces to studying the behavior of the solution $\mu_t^\beta$ to the continuity equation (2) as $\beta \to \infty$. The PDE analysis in this limit will therefore be the focus of the following sections.

## 3.2 The Alignment Phase

To characterize the limiting dynamics in the first phase we make the following assumptions:

**Assumption 1.** $Q, K, V$ *are invertible square matrices.*

**Assumption 2.** *The probability measure $\mu_0$ on $S^{d-1}$ is absolutely continuous with respect to the Lebesgue measure on $\mathbb{S}^{d-1}$. Its density is bounded from above and below ($\min_{x \in \mathbb{S}^{d-1}} \mu_0(x) > 0$) and Lipschitz continuous.*

These technical assumptions, significantly milder than the ones made in most related works, are needed to guarantee that the terms appearing in the analysis of the limiting equation, e.g., the denominator in (5), are sufficiently well behaved. Under these conditions, we show that the limiting dynamics in this regime coincides with the formal Laplace approximation of term (I) in (4), i.e., the integrals in the definition of the vector fields can be replaced by the value of the integrand at the maximum point $x' = K^T Q x / |K^T Q x|$, leading to the significantly simplified expression (5) below.

**Theorem 3.2.** *Let Assumptions 1, 2 hold, then the solutions $\{\mu_\beta\}_\beta$ of the continuity equation (2) converge in $\mathcal{C}([0,T], \mathcal{P}(\mathbb{S}^{d-1}))$ to the solution $\mu^\infty$ of the partial differential equation:*

$$\begin{cases} \partial_t \mu(x) &= -\operatorname{div}\left(\mu(x) P_x \frac{V K^T Q x}{|K^T Q x|}\right), \\ \mu(0, x) &= \mu_0(x), \quad x \in \mathbb{S}^{d-1}. \end{cases} \tag{5}$$

*Proof Sketch.* To establish the result we must prove well-posedness of the family of equations leading to the desired limit and obtain sufficient regularity uniformly in $\beta$ to ensure that the formal simplifications from (4)(I) to (5) are allowed. This is particularly important as the derivatives of the kernel tend to infinity in the limit $\beta \to \infty$. The core argument proceeds in three main steps. First, we establish the relative compactness of the family of trajectories $\{\mu^\beta\}_{\beta>0}$ in the space $\mathcal{C}([0,T], \mathcal{P}(\mathbb{S}^{d-1}))$ using a variant of Ascoli-Arzelà theorem and the boundedness of the vector field $\chi_\beta[\mu^\beta]$. The second, crucial step involves deriving uniform in $\beta$ estimates on the regularity (i.e., Lipschitz bounds) of the vector field $\chi_\beta[\mu^\beta]$ along the solution trajectories $\mu^\beta$. This is achieved by analyzing the concentration behavior of the kernel $e^{\beta\langle Qx, Ky\rangle}$ as $\beta \to \infty$, leveraging properties related to the cumulants of the Von Mises-Fisher distribution, and employing a continuation argument to propagate regularity over time. Finally, using the compactness and uniform regularity, we pass to the limit $\beta \to \infty$ in the weak formulation of the continuity equation (2). The uniform estimates allow us to conclude that $\mu^\infty$ is a solution of (5), while uniqueness follows from [5]. The full proof is deferred to Appendix A.1. □

A consequence of Theorem 3.2 is that in the large-$\beta$ limit, the tokens, to leading order, evolve independently of each other, driven primarily by the structure of the $Q$, $K$, and $V$ matrices. In this regime, self-attention behaves like a composition of linear layers followed by layer normalization, with minimal influence from inter-token interactions.

Combining the above result with Lemma 3.1 we obtain the following convergence result:

**Corollary 3.3.** *Under Assumptions 1, 2, for every $t > 0$ we have $W_1(\mu_t^{N,\beta}, \mu_t^\infty) \to 0$ as $N \to \infty$, provided that $\beta_N \to \infty$ slowly enough.*

Having established that $\mu^\infty$ is a solution of equation (5), we can investigate its long-time behavior. In particular, we show below that the support of $\mu_t^\infty$ is asymptotically flattened onto a lower-dimensional subspace determined by the spectral properties of the matrix $V K^\top Q$,

**Proposition 3.4.** *Let $\mu_0$ be a probability measure on $\mathbb{S}^{d-1}$ absolutely continuous with respect to the Lebesegue measure, and let $\mu_t$ be the corresponding solution of (5). Then for every $\nu \in \omega(\mu_0)$ (the $\omega$-limit set of $\mu_0$) it holds:*

$$supp(\nu) \subseteq E_{max} \cap \mathbb{S}^{d-1},$$

*where $E_{max}$ is the generalized eigenspace associated to the eigenvalue of $VK^TQ$ with largest real part.*

*Proof.* The proof is provided in Appendix A.2, where we reduce the analysis to a linear system of ODEs in $\mathbb{R}^d$ with matrix $VK^TQ$, identifying the corresponding asymptotics with the ones of (5). □

**Remark 3.5.** *At a first glance, this result might appear inconsistent with those of [11], since in some cases measures supported on $E_{\max}$ do not maximize the energy. However, this apparent discrepancy is a consequence of the order of the limits being taken, with $\beta \to \infty$ preceding $t \to \infty$ in our case.*

Proposition 3.4 demonstrates that the token representations rapidly collapse onto a lower-dimensional subspace determined by the model's matrices. This can be interpreted as the initial phase of the inference process, where information is compressed into a smaller, more relevant subspace. This phenomenon is consistent with the rank collapse observed, e.g., in [38, 27].

**Remark 3.6.** *Apart from the collapse to $E_{\max}$, one cannot in general conclude the existence of a limiting (stationary) dynamics for (5). Indeed, it is not difficult to construct examples where the particles continue to rotate on the sphere indefinitely, e.g., when $V$ is a rotation and $Q^T K = \mathrm{Id}$.*

**Remark 3.7.** *Recent works have studied transformer models with stochastic perturbations [48], where the token dynamics is influenced by random noise. In this setting, the convergence to the corresponding equation (5) (with an additional Laplacian term) is typically easier to establish due to the regularizing effect of the noise (see [39]).*

### 3.3 The Heat Phase

Having established the rapid collapse onto the subspace $E_{max} \cap \mathbb{S}^{d-1}$, we now investigate the slower evolution within this subspace, assuming that the initial measure $\mu_0$ is supported in $E_{\max} \cap \mathbb{S}^{d-1}$ as a consequence of the previous analysis:

**Assumption 3.** *The initial condition $\mu_0$ in the heat phase satisfies $supp(\mu_0) \subseteq E_{max} \cap \mathbb{S}^{d-1}$.*

Since the intersection $E_{max} \cap \mathbb{S}^{d-1}$ can be identified with a lower-dimensional sphere, specifically $\mathbb{S}^{\dim(E_{\max})-1}$, we will, with a slight abuse of notation, continue to denote it by $\mathbb{S}^{d-1}$.

To demonstrate that the heat equation, described in a different setting in [46], emerges as an intermediate dynamical phase due to the spherical geometry induced by LayerNorm, we assume:

**Assumption 4.** $Q^T K|_{E_{max}} = \lambda_1 I$ and $V|_{E_{max}} = \pm\lambda_2 I$ *when restricted to $E_{max}$, with $\lambda_1, \lambda_2 > 0$.*

Under this condition, $E_{max}$ is an invariant subspace for Eq. (2) and, without loss of generality, we can suppose $\lambda_1, \lambda_2 = 1$.

**Remark 3.8.** *Assumption 4, for example, is satisfied under the global assumption $Q^\top K = S$ and $V = \pm S$, with $S$ symmetric definite positive matrix. This is a fairly standard assumption in recent studies within this framework and it endows the model with an additional structure of gradient flow on $\mathcal{P}(\mathbb{S}^{d-1})$ with respect to a modified metric (see [11, 25]).*

In this regime, the vector field $\chi^\beta[\mu]$ vanishes on the support of $\mu$ as $\beta \to \infty$, but its rescaled version admits the formal limit (see Corollary B.2):

$$\beta\chi_\beta[\mu](x) \to \gamma\frac{\nabla_{\mathbb{S}^{d-1}}\mu}{\mu}(x), \quad \text{as } \beta \to \infty,$$

where $\gamma := \pm 1$, depending on the sign choice in the definition of $V$. This scaling of the vector field by $\beta$ corresponds to a time rescaling $dt = \beta ds$, explaining the phase duration of order $O(\beta)$.

**Proposition 3.9.** *Let Assumption 4 hold and let $\mu_0^\infty \in \mathcal{P}(\mathbb{S}^{d-1})$ be the initial measure. Assume that there exist $T > 0$, $k$ positive integer, and $\mu_t^\infty \in C([0,T], C^{k+3}(\mathbb{S}^{d-1}))$, with $\min_{x\in\mathbb{S}^{d-1}} \mu_t^\infty(x) > 0$ for all $t \in [0,T]$, such that $\mu_t^\infty$ solves the heat equation on $[0,T] \times \mathbb{S}^{d-1}$:*

$$\begin{cases} \partial_t\mu & = -\gamma\Delta\mu, \\ \mu(0) & = \mu_0, \end{cases} \tag{6}$$

where $\Delta$ denotes the Laplace-Beltrami operator on $\mathbb{S}^{d-1}$. Then, for large $\beta$, $\mu_t^\infty$ solves the mean-field PDE:

$$\begin{cases} \partial_t \mu & = -\mathrm{div}(\mu\,\beta\chi_\beta[\mu]) + R_\beta, \\ \mu(0) & = \mu_0, \end{cases}$$

where the residual term satisfies $R_\beta \to 0$ in $C([0,T], C^k(\mathbb{S}^{d-1}))$ as $\beta \to \infty$.

This proposition, whose proof is provided in Appendix B, characterizes the limiting dynamics within the lower-dimensional manifold, connecting the transformer model with a heat flow on the sphere, thereby justifying the name of this phase. Remarkably, this connection holds without the need for correction terms, in contrast to [46]. We now need to distinguish between two different cases:

- **Forward diffusion**. When $\gamma < 0$ in equation (6), the dynamics corresponds to a forward heat equation. In this setting, local existence and regularity for $\mu_t^\infty$ (and in particular the assumptions of Prop. 3.9) are automatically satisfied due to the smoothing properties of forward diffusion, provided that $\mu_0 \in C^{k+3}(\mathbb{S}^{d-1})$. Notably, interacting particle systems of the specific form given by equation (SA) (under the assumption $Q^\top K = \mathrm{Id} = -V$) have been studied in the literature and are known as *diffusion-velocity methods*; see, for instance, [33, 20, 8, 31, 32, 35].
- **Backward diffusion**. When $\gamma > 0$, the dynamics corresponds to a backward heat equation. In this case, the regularity assumptions on $\mu_0^\infty$ ensuring local existence and regularity are significantly more restrictive (e.g. requiring that $\mu_0$ is in the Gevrey-$\frac{1}{2}$ space). Nonetheless, we construct explicit examples of solutions below. The backward heat equation is a prototypical ill-posed problem, which explains why the statement of Proposition 3.9 is necessarily weaker than that of Theorem 3.2.

A family of initial conditions $\mu_0$ that satisfies the assumptions of Proposition 3.9 is given by

$$\mu_0 = \sum_{j=1}^{M} \alpha_j \mathcal{N}_{\mathbb{S}^{d-1}}(m_j, \sigma_j^2),$$

where $\alpha_j \geq 0$, $\sum_{j=1}^{M} \alpha_j = 1$, and $\mathcal{N}_{\mathbb{S}^{d-1}}(m, \sigma^2)$ denotes the heat kernel (the forward-in-time evolution under the heat semi-group $\exp(t\Delta)$ of a Dirac delta, analogous to a Gaussian $\mathcal{N}_{\mathbb{R}^d}(m, \sigma^2)$ in Euclidean space) centered at $m \in \mathbb{S}^{d-1}$ with concentration related to $\sigma^2$. By linearity of $\Delta$, the explicit solution to $\partial_t \mu = -\gamma \Delta \mu$ is then given by:

$$\mu_t^\infty = \sum_{j=1}^{M} \alpha_j \mathcal{N}(m_j, \sigma_j^2 - \gamma t). \tag{7}$$

For forward diffusion ($\gamma < 0$), this solution is a smooth function for all $t \geq 0$. while in the backward case ($\gamma > 0$) this only holds for $t \in [0, T_{\min})$, where $T_{min} = \min_j \sigma_j^2$ is the time at which the first Gaussian component collapses to a Dirac delta $\delta_{m_j}$. A more general class in which local existence and well-posedness hold in both the forward and backward directions is the set of positive, Gevrey-1/2, functions.

Motivated by the aggregation behavior observed in the finite-$\beta$ particle system, we conjecture that the collapsed $\delta_{m_j}$ remains invariant under the limiting dynamics, while other components continue evolving independently according to the backward heat equation until their respective collapse times. From the practical perspective, this observation suggests that the transformer's behavior in this regime can be interpreted as a form of regularized denoising (when $\beta$ is finite) acting on the input. This aligns with the clustering phenomena extensively studied in previous works on the model. The dynamics in this phase, governed by a heat equation, drive the formation of distinct token clusters (via backward diffusion) or the smoothing of the token distribution (via forward diffusion). This behavior can be interpreted as a representation refinement stage, where tokens are organized into more defined semantic groups.

**Remark 3.10.** *In [25], a simplified model, referred to as the Unnormalized Self-Attention (USA) model, is proposed, where the normalization factor $Z_{\beta,\mu}(x)$ is replaced by a constant $Z_\beta$, significantly simplifying the mathematical analysis. By choosing $Z_\beta = \frac{1}{\beta}\int_{\mathbb{S}^{d-1}} e^{\beta\langle x,y\rangle}\,d\sigma(y)$ (or equivalently by rescaling time), the limiting behavior of the model no longer yields the heat equation, but rather the porous medium equation: $\partial_t \mu = \Delta(\mu^2)$. Even in this case, the convergence of particles system to this nonlinear PDE has been extensively studied (see for example [22, 33, 39, 41] or [47] for $\mathbb{S}^{d-1}$).*

### 3.4 Pairing Phase

The initial conditions we consider for the dynamics on longer timescales must be compatible with the steady states of the preceding phase. Motivated by the discussion at the end of the previous section, we therefore formulate the following assumption:

**Assumption 5.** *The initial condition $\mu_0$ in the pairing phase can be written as $\mu_0 = \sum_{j=1}^{m} \alpha_j \delta_{x_j}$ for an $m \in \mathbb{N}$, with $x_j \in \mathbb{S}^{d-1}$, $\alpha_j > 0$ $\forall j \in \{1, ..., m\}$ and $\sum_{j=1}^{m} \alpha_j = 1$.*

Under this assumption, further supposing for the sake of clarity that $\alpha_j = 1/m$ for all $j \in \{1, \ldots, m\}$, we can interpret each cluster as a particle, and the dynamics of the system is given by the set of ODEs (SA). In the regime of large $\beta$, clusters interact very weakly due to their separation and the exponential tails (in $\beta$) of the interaction kernel, resulting in exponentially long timescales for the nontrivial dynamics. Here, analogously to [3], interactions are dominated by the closest pair of clusters $(\underline{i}, \underline{j})$, assumed unique, satisfying $\langle x_{\underline{i}}, x_{\underline{j}} \rangle = \max_{i \neq j} \langle x_i, x_j \rangle$ at initialization. We note that this hardmax particle interaction, as well as the timescale where it arises in the large $\beta$ limit, was introduced in [23, Section 6] in the case $d = 2$. We present an analogous result here in arbitrary dimension, without claiming originality, to provide a complete dynamical picture across phases.

**Proposition 3.11.** *The solutions $x_i(t)$ of the ODE system* (SA)*, under Assumptions 4 and positive $V$, with the rescaled time $dt = e^{\beta(1 - \langle x_{\underline{i}}, x_{\underline{j}} \rangle)} ds$, converge as $\beta \to \infty$ to the solutions of the system:*

$$\begin{cases} \dot{y}_k(t) = \begin{cases} P_{y_{\underline{i}}}(y_{\underline{j}}) & \text{if } k = \underline{i}, \\ P_{y_{\underline{j}}}(y_{\underline{i}}) & \text{if } k = \underline{j}, \\ 0 & \text{otherwise}, \end{cases} \\ y_i(0) = x_i(0) \end{cases}$$

*on finite intervals $[0, T_\epsilon]$, with $T_\epsilon$ such that $\langle y_{\underline{i}}, y_{\underline{j}} \rangle \leq 1 - \epsilon$ throughout the interval, for any $\epsilon > 0$.*

In other words, all clusters remain stationary except for the closest pair, which collapses along the geodesic connecting them, in a time exponential in $\beta$. Note that this result only holds up to an arbitrary moment before the first collapse. We refer to [23, Section 6] for a detailed explanation of the challenges to bypass this limit and a proof of an analogous result until and beyond the collapse time in a related but simplified model. The above proposition is proven for completeness in Appendix C.

This final, slow phase models the sequential merging of the closest token clusters. This can be interpreted as the construction of higher-order abstractions, where previously formed groups are hierarchically combined to create more complex representations.

## 4 Numerical experiments

This section presents numerical simulations of the transformer model in Eq. (1). All experiments are conducted in dimension $d = 3$ or or $d = 2$ to facilitate visualization and are designed to validate our theoretical findings. The attention mechanism is implemented using the official PyTorch function `torch.nn.functional.scaled_dot_product_attention()` and the experiments are performed on a single Nvidia H100. The code is available at [28].

**First Phase Dynamics.** Figure 1 illustrates the dynamics of the alignment phase, showing distinct behaviors based on the parameters choices for $Q, K, V$. For both scenarios presented in Figure 1, the initial state consists of $N = 10^4$ tokens sampled independently and identically uniformly from the sphere $\mathbb{S}^2$. We set the inverse temperature parameter $\beta = 30$ and use a time step of $dt = 10^{-2}$.

- **Scenario 1a (collapse to 1D subspace).** The matrix $VK^TQ$ is chosen such that it possesses a unique eigenvalue with maximal real part. As predicted by our theory, this configuration leads to the tokens collapsing onto a one-dimensional subspace (i.e. two antipodal points).

- **Scenario 1b (non-gradient flow dynamics and rotation).** This example employs a parameter choice for $Q, K, V$ that falls outside the gradient flow regime. Nevertheless, the tokens are observed to collapse toward a two-dimensional subspace (a great circle), accompanied by a collective rotation of the particles along this circle.

Both observed behaviors are consistent with the results in Theorem 3.2 and Proposition 3.4.

$$V = \begin{bmatrix} 1 & 0 & 0 \\ 0 & -1 & 0 \\ 0 & 0 & 2 \end{bmatrix}$$

$$Q^T K = \begin{bmatrix} -1 & 0 & 0 \\ 0 & -1 & 0 \\ 0 & 0 & 1 \end{bmatrix}$$

$$V = \begin{bmatrix} 1 & 0 & 0 \\ 0 & 1 & 0 \\ 0 & 0 & 1 \end{bmatrix}$$

$$Q^T K = \begin{bmatrix} 0 & 1 & 0 \\ -1 & 0 & 0 \\ 0 & 0 & -2 \end{bmatrix}$$

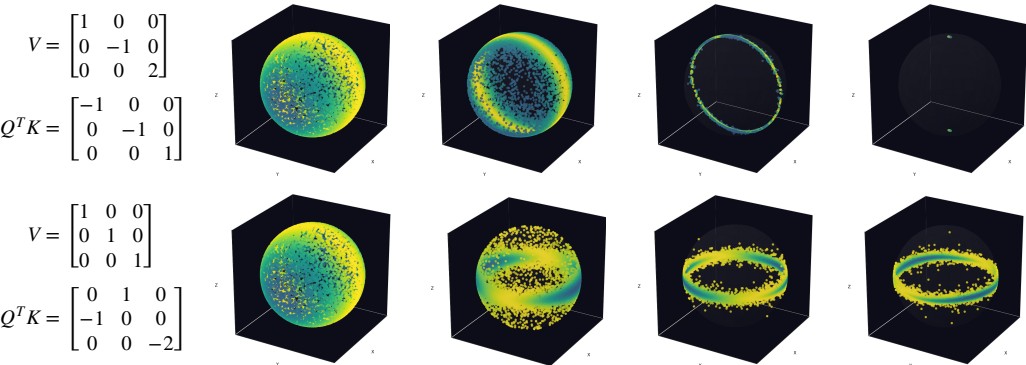

Figure 1: Simulations of two different scenarios (one per row, four timesteps) for the first phase.

**Second Phase Dynamics.** We support the conclusions of section 3.3 through two examples, comparing empirical dynamics with analytical solutions of backward and forward diffusion equations.

- **Scenario 2a (collapse to 2D subspace and backward diffusion).** For the experiment in Figure 2 we set $\beta = 10$. The initial configuration comprises $N = 10^4$ i.i.d. tokens. Their elevation angle $\psi$ is sampled uniformly on $[-\frac{\pi}{2}, \frac{\pi}{2}]$, while their azimuthal angles, $\theta_i \in [0, 2\pi)$, are distributed according to the mixture density $g(\theta)$:

$$g(\theta) := 0.2 \cdot \mathcal{N}(\theta; \pi/2, \sigma_0) + 0.5 \cdot \mathcal{N}(\theta; 0, \sigma_0) + 0.3 \cdot \mathcal{N}(\theta; 4\pi/3, \sigma_0)$$

  where $\mathcal{N}(\cdot; \mu, \sigma_0)$ denotes the probability density function of a wrapped normal distribution on $\mathbb{S}^1$ with mean $\mu$ and standard deviation $\sigma_0 = 0.2$. The parameters $Q$, $K$, and $V$ are chosen so that, after the first phase, the tokens collapse onto the $xy$-plane, with distribution $g(\theta)$. The analytical solution to the backward heat equation with initial condition $g(\theta)$ (computed as in Eq. (7)) is plotted as a red curve in Figure 2. The positions of the clusters agree with this solution, numerically confirming the predictions of Proposition 3.9.

- **Scenario 2b (forward diffusion comparison).** In Figure 3, we compare the empirical token distribution with the analytical solution of the forward heat equation characterizing a possible example of the second phase of the dynamics. Specifically, we simulate the evolution of $5 \times 10^4$ tokens, initially sampled from a superposition of three Gaussian densities on $\mathbb{S}^1$, through the transformer model with parameters $\beta = 50$, $d = 2$, $Q = K = \text{Id}$, $V = -\text{Id}$, and $dt = 10^{-3}$. The analytical solution of the forward heat equation (in red) closely matches the token distribution histogram (in blue) over time (i.e., depth). Note that, as expected, the forward diffusion process is significantly more stable numerically than the backward one.

$$V = \begin{bmatrix} 1 & 0 & 0 \\ 0 & 1 & 0 \\ 0 & 0 & 0.1 \end{bmatrix}$$

$$Q^T K = \begin{bmatrix} 1 & 0 & 0 \\ 0 & 1 & 0 \\ 0 & 0 & 0.1 \end{bmatrix}$$

Analytical solution

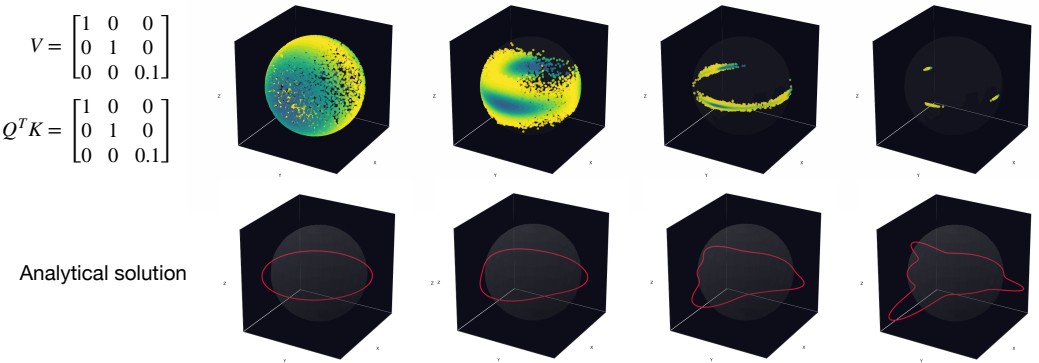

Figure 2: Numerical simulation of the backward scenario for the second phase on $\mathbb{S}^2$.

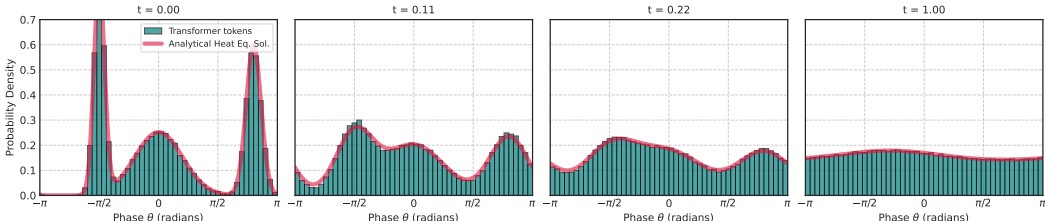

Figure 3: Evolution of the tokens distribution in the forward scenario for the second phase in $\mathbb{S}^1$.

## 5 Conclusions

This work provides a mathematical analysis of token dynamics in mean-field transformer models within the moderate interaction regime, where interaction strength ($\beta$) scales with context size ($N$), motivated by scaling practices in modern LLMs. Our study reveals a fundamental multiscale structure governing the evolution of token representations through network depth in this setting. More specifically, we showed that, under this scaling, the system progresses through a sequence of three different dynamical phases characterized by qualitatively distinct dynamical behavior, operating on separated timescales. Through our analysis, we offer a unified dynamical picture describing how deep transformers might achieve progressive representation refinement.

This unified dynamical picture is, however, not yet fully rigorous. Indeed, while we establish convergence results for the alignment phase and provide partial justification for the intermediate and late phases under specific assumptions, a complete mathematical treatment of the full dynamics - particularly of the backward heat regime beyond the first collapse and of the slow clustering interactions - remains an open challenge due to significant technical difficulties in the analysis of the strongly unstable limiting equations. Additionally, characterizing phase transitions in self-attention for different $N, \beta$ scalings is an interesting separate question, with progress made in [15] under assumptions on inter-token angles.

Furthermore, while the first dynamical phase has quite general assumptions on the parameter matrices, the following phases still require relatively limited assumptions (although less limited than in most previous works). Relaxing these assumptions further, in particular in the case of non-gradient dynamics, would constitute an interesting, but also technically quite challenging, avenue of future research.

There are several important directions in which our work could be extended. Most notably, incorporating the MLP, which could be interpreted as introducing a drift term in the dynamics, acting independently on each token without accounting for mutual interactions. Another natural extension involves studying the dynamics under more general parameter settings. For instance, during the heat phase, we assume that $Q^T K = S$ is symmetric positive definite, which holds, for example, when $Q = K$. This "shared-QK" assumption is not novel and has been adopted in prior empirical work (e.g., in [30]). While different choices of these parameters (both MLP and Attention matrices) can have a dramatic effect on model behavior, with adversarial choices potentially leading to qualitatively different dynamics from the one predicted in this paper, we believe our results to be a relevant first step towards understanding the development of representations in transformers, capturing some important qualitative features of these models as shown in [25].

A further direction of future research consists in providing sufficient conditions for the stability of the space $E_{max}$ emerging in the alignment phase under the prelimit model (i.e., for large but finite $\beta$), thereby justifying Assumption 3 and, ultimately, connecting in a rigorous way the alignment and heat phases identified in this paper.

While the path from this theoretical analysis to direct application is not immediate, we believe our work opens several potential avenues for future investigation. The characterization of the alignment phase, for instance, offers a potential mechanism for interpreting how token representations evolve into learned subspaces. Finally, by focusing on the "moderate interaction regime", we hope our analysis provides a theoretical foundation for a more principled understanding of parameter scaling, particularly as models are adapted for longer contexts.

## Acknowledgements

AA and GB acknowledge partial support by the Institute of Mathematical Statistics and Actuarial Sciences at the University of Bern. FP acknowledges support by the American Mathematical Society through the Bergman fellowship. AA thanks B. Geshkovski, J. Mattingly, Y. Polyanskiy, and P. Rigollet for interesting discussions. All authors thank the anonymous reviewers for comments that have improved the quality of the paper. Calculations were performed on UBELIX, the HPC cluster at the University of Bern.

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

# A Proofs of the alignment phase

This section is divided into two parts. The first part contains the proof of Theorem 3.2, while the second one contains the proof of Proposition 3.4.

## A.1 Moderate scaling limit

Consider the family $\{\mu^\beta\}_{\beta \geq 0}$ of solutions to the usual continuity equation (2):

$$\begin{cases} \partial_t \mu^\beta & = -div(\mu^\beta \chi_\beta[\mu^\beta]), \\ \mu^\beta(0,x) & = \mu_0(x), \quad x \in \mathbb{S}^{d-1}, \end{cases}$$

where $\chi_\beta$ is the vector field given by:

$$\chi_\beta[\mu](x) = \frac{\int_{S^{d-1}} e^{\beta\langle x, By\rangle} P_x V y \mu(dy)}{\int_{S^{d-1}} e^{\beta\langle x, By\rangle} \mu(dy)}.$$

**Remark A.1.** *For notational simplicity, we will refer to the matrix $Q^t K$ in the main body as $B$.*

**Remark A.2.** *In the following $C$ will be a constant depending only on $V, B, d$ and $\mu_0$. Its value may change line by line.*

Under assumptions 1, 2, i.e.:

- A1: $Q, B$ are invertible square matrices.

- A2: The probability measure $\mu_0$ on $S^{d-1}$ is absolutely continuous with respect to the Lebesgue measure. Its density is bounded and Lipschitz continuous and its minimum satisfies $\min_{x \in \mathbb{S}^{d-1}} \mu_0(x) > 0$.

one can prove Theorem 3.2: $\mu_\beta$ converges weakly to $\mu_\infty$ in $C([0,T]; \mathcal{P}(S^{d-1}))$ where $\mu_\infty$ is the unique solution of the PDE:

$$\begin{cases} \partial_t \mu & = -div(\mu \frac{P_x V B^T x}{|B^T x|}), \\ \mu(0,x) & = \mu_0(x), \quad x \in \mathbb{S}^{d-1}. \end{cases}$$

and the metric in $\mathcal{C}([0,T], \mathcal{P}(\mathbb{S}^{d-1}))$ is given by:

$$d(\mu, \nu) := \sup_{t \in [0,T]} \rho(\mu_t, \nu_t) = \sup_{t \in [0,T]} \sup_{f \in BL(\mathbb{S}^{d-1})} \left| \int_{\mathbb{S}^{d-1}} f(x)\mu_t(dx) - f(x)\nu_t(dx) \right|,$$

where $BL(\mathbb{S}^{d-1})$ is the set of all the Lipschitz continuous functions on $\mathbb{S}^{d-1}$ which are bounded together with their Lipschitz constant by 1.

The idea of the proof follows five steps:

- Relative compactness in $C([0,T]; \mathcal{P}(S^{d-1}))$,

- Bounds on cumulants of the Von Mises-Fisher distribution,

- Prove a relationship between the derivatives of $\mu$ and the regularity of the vector field,

- Apply a continuation argument to show the uniform regularity of $\chi_\beta[\mu_\beta]$,

- Use the regularity to pass to the limit in the PDE.

### A.1.1 Relative compactness

**Proposition A.3.** *The set $\{\mu_\beta\}_{\beta \geq 0}$ is relatively compact in $\mathcal{C}([0,T], \mathcal{P}(\mathbb{S}^{d-1}))$.*

*Proof.* By Prokhorov's theorem, since $\mathbb{S}^{d-1}$ is compact, we can conclude that $\mathcal{P}(\mathbb{S}^{d-1})$ is weakly compact. Since $\rho$ metricizes the weak topology, then also $(\mathcal{P}(\mathbb{S}^{d-1}), \rho)$ is compact. To apply Ascoli-Arzelà theorem, we just need the equicontinuity of the set $\{\mu^\beta\}_{\beta > 0}$.

Given $0 \leq s \leq t \leq T$ and $\beta > 0$:

$$\rho(\mu_s^\beta, \mu_t^\beta) = \sup_{f \in BL(\mathbb{S}^{d-1})} \left| \int_{\mathbb{S}^{d-1}} f(x)(\mu_t^\beta(dx) - \mu_s^\beta(dx)) \right|$$

$$\leq \sup_{f \in BL(\mathbb{S}^{d-1})} \int_s^t \left| \int_{\mathbb{S}^{d-1}} \langle \nabla f(x), \chi^\beta[\mu_u^\beta](x) \rangle \mu_u^\beta(dx) du \right|$$

$$\leq \int_s^t \int_{\mathbb{S}^{d-1}} |\chi^\beta[\mu_u^\beta](x)| \mu_u^\beta(dx) du \leq |t - s|.$$

This is sufficient to conclude the proof. $\qquad\square$

### A.1.2 Bounds on the vector field

The aim of the following paragraphs is to obtain some bounds on $D^i \chi_\beta[\mu]$, $i = 0, 1, 2$.

To fix the notation we define the probability measure $\nu_x^{\mu,\beta,B}$ on $\mathbb{S}^{d-1}$ as:

$$\nu_x^{\mu,\beta,B}(dy) := \frac{e^{\beta\langle x, By \rangle} \mu(dy)}{\int_{S^{d-1}} e^{\beta\langle x, By \rangle} \mu(dy)}.$$

**Remark A.4.** *The measure $\nu_x^{\sigma,\beta,B}$ is the Von Mises-Fisher distribution with mean direction $\frac{B^T x}{|B^T x|}$ and concentration parameter $\beta|B^T x|$. Some properties of this distribution are studied later.*

Then the vector field $\chi_\beta[\mu]$ can be written as:

$$\chi_\beta[\mu](x) = P_x V \left( \mathbb{E}_{\nu_x^{\mu,\beta,B}}[Y] \right) = P_x V \left( \mathbb{E}_{\nu_x^{\mu,1,Id}}[Y] \right) \circ (\beta B^T x),$$

where $\circ$ denotes the composition with respect to the parameter $x$ of the measure $\nu_x^{\mu,\beta,B}$.

**Lemma A.5.** *The derivatives of the vector field $\chi_\beta[\mu]$ are bounded by:*

$$|\chi_\beta[\mu](x)| \leq C,$$

$$|D_x^1 \chi_\beta[\mu](x)| \leq C \left( 1 + \beta \left| \mathbb{E}_{\nu_x^{\mu,\beta,B}}\left[ \left( Y - \mathbb{E}_{\nu_x^{\mu,\beta,B}}[Y] \right)^{\otimes 2} \right] \right| \right),$$

$$|D_x^2 \chi_\beta[\mu](x)| \leq C \left( 1 + \beta \left| \mathbb{E}_{\nu_x^{\mu,\beta,B}}\left[ \left( Y - \mathbb{E}_{\nu_x^{\mu,\beta,B}}[Y] \right)^{\otimes 2} \right] \right| + \beta^2 \left| \mathbb{E}_{\nu_x^{\mu,\beta,B}}\left[ \left( Y - \mathbb{E}_{\nu_x^{\mu,\beta,B}}[Y] \right)^{\otimes 3} \right] \right| \right)$$

*where $C$ is a constant depending only on $V, B, d$.*

*Proof.* Let's compute the derivatives of $\chi_\beta[\mu]$:

$$|\chi_\beta[\mu](x)| \leq |P_x V| |\mathbb{E}_{\nu_x^{\mu,\beta,B}}[Y]| \leq C_V,$$

$$|D_x^1 \chi_\beta[\mu](x)| \leq |D_x^1 P_x V| \left| \left( \mathbb{E}_{\nu_x^{\mu,1,Id}}[Y] \circ (\beta B^T x) \right) \right| + |P_x V| |D_x^1 \left( \mathbb{E}_{\nu_x^{\mu,\beta,B}}[Y] \right)|$$

$$\leq C_V \left( 1 + |D_x^1 \left( \mathbb{E}_{\nu_x^{\mu,\beta,B}}[Y] \right)| \right),$$

$$|D_x^2 \chi_\beta[\mu](x)| \leq |D_x^2 P_x V| |\mathbb{E}_{\nu_x^{\mu,\beta,B}}[Y]| + 2|D_x^1 P_x V| |D_x^1 \left( \mathbb{E}_{\nu_x^{\mu,\beta,B}}[Y] \right)| + |P_x V| |D_x^2 \left( \mathbb{E}_{\nu_x^{\mu,\beta,B}}[Y] \right)|$$

$$\leq C_V (1 + 2|D_x^1 \left( \mathbb{E}_{\nu_x^{\mu,\beta,B}}[Y] \right)| + |D_x^2 \left( \mathbb{E}_{\nu_x^{\mu,\beta,B}}[Y] \right)|$$

Hence we need to compute the derivatives with respect to $x$ of $\mathbb{E}_{\nu_x^{\mu,\beta,B}}[Y] = \mathbb{E}_{\nu_x^{\mu,1,Id}}[Y] \circ (\beta B^T x)$. Thanks to the Faa di Bruno formula:

$$D_x^n \mathbb{E}_{\nu_x^{\mu,\beta,B}}[Y] = \sum_{\pi \in \Pi_n} \left( (D_x^{|\pi|} \mathbb{E}_{\nu_x^{\mu,1,Id}}[Y])|_{\beta B^T x} \circ \bigotimes_{P \in \pi} D^P (\beta B^T x) \right),$$

with $\Pi_n$ the set of all the possible partitions of $\{1, ..., n\}$ and $\otimes$ the tensor product. The previous expression can be bounded by $C_B \sum_{l=0}^n \beta^l \|(D_x^l \mathbb{E}_{\nu_x^{\mu,1,Id}}[Y])|_{\beta B^T x}\|$, where $C_B$ is a constant depending on the matrix $B$.

Thus, the aim is to compute a bound for $D_x^n \mathbb{E}_{\nu_x^{\mu,1,Id}}[Y]$. This is related to the $d-$ dimensional cumulants (tensors) of the distribution $\nu_x^{\mu,1,Id}$. Indeed, we can write:

$$
\begin{aligned}
D_x^n \left( \mathbb{E}_{\nu_x^{\mu,1,Id}}[Y] \right)_x &= D_v^n \left( \mathbb{E}_{\nu_{x+v}^{\mu,1,Id}}[Y] \right)_{v=0} \\
&= D_v^n \left( \frac{\int e^{\langle x+v,y \rangle} y\, \mu(dy)}{\int e^{\langle x+v,y \rangle} \mu(dy)} \right)_{v=0} \\
&= D_v^n \left( \frac{\int e^{\langle v,y \rangle} y\, e^{\langle x,y \rangle} \mu(dy)}{\int e^{\langle x,y \rangle} \mu(dy)} \frac{\int e^{\langle x,y \rangle} \mu(dy)}{\int e^{\langle v,y \rangle} e^{\langle x,y \rangle} \mu(dy)} \right)_{v=0} \\
&= D_v^n \left( \frac{\int e^{\langle v,y \rangle} y\, \nu_x^{\mu,1,Id}(dy)}{\int e^{\langle v,y \rangle} \nu_x^{\mu,1,Id}(dy)} \right)_{v=0} \\
&= D_v^{n+1} \left( \log \mathbb{E}_{\nu_x^{\mu,1,Id}}[e^{\langle v,Y \rangle}] \right)_{v=0}.
\end{aligned}
\tag{8}
$$

It is well known that the first three cumulants correspond to the central moments:

$$
\begin{aligned}
D_x^0 \left( \mathbb{E}_{\nu_x^{\mu,1,Id}}[Y] \right)_x &= \mathbb{E}_{\nu_x^{\mu,1,Id}}[Y], \\
D_x^1 \left( \mathbb{E}_{\nu_x^{\mu,1,Id}}[Y] \right)_x &= \mathbb{E}_{\nu_x^{\mu,1,Id}}\left[ \left( Y - \mathbb{E}_{\nu_x^{\mu,1,Id}}[Y] \right)^{\otimes 2} \right], \\
D_x^2 \left( \mathbb{E}_{\nu_x^{\mu,1,Id}}[Y] \right)_x &= \mathbb{E}_{\nu_x^{\mu,1,Id}}\left[ \left( Y - \mathbb{E}_{\nu_x^{\mu,1,Id}}[Y] \right)^{\otimes 3} \right].
\end{aligned}
\tag{9}
$$

The thesis then follows by replacing these equalities in the initial bounds (after renaming $C_V$ and $C_B$). $\qquad\square$

**Lemma A.6.** *Let $\sigma$ be the uniform measure on $\mathbb{S}^{d-1}$. Then:*

$$
|\mathbb{E}_{\nu_x^{\sigma,\beta,Id}}[Y]| \leq 1,
$$

$$
|\mathbb{E}_{\nu_x^{\sigma,\beta,Id}}\left[ \left( Y - \mathbb{E}_{\nu_x^{\sigma,\beta,Id}}[Y] \right)^{\otimes 2} \right]| \leq C\frac{1}{\beta},
$$

$$
|\mathbb{E}_{\nu_x^{\sigma,\beta,Id}}\left[ \left( Y - \mathbb{E}_{\nu_x^{\sigma,\beta,Id}}[Y] \right)^{\otimes 3} \right]| \leq C\frac{1}{\beta^2}.
$$

*Proof.* By (8) and Schwarz's theorem the three tensors are invariant by permutations of the indices and by definition of $\nu_x^{\sigma,\beta,Id}$ they are also invariant by rotations that fix $x$. Hence (see lemma D.2) they must have the form:

$$
\mathbb{E}_{\nu_x^{\sigma,\beta,Id}}[Y] = \alpha_1 x,
$$

$$
\mathbb{E}_{\nu_x^{\sigma,\beta,Id}}\left[ \left( Y - \mathbb{E}_{\nu_x^{\sigma,\beta,Id}}[Y] \right)^{\otimes 2} \right] = D_x^1 \left( \mathbb{E}_{\nu_x^{\sigma,1,Id}}[Y] \right)_{\beta x} = \alpha_2 x \otimes x + \beta_2 I,
\tag{10}
$$

$$
\mathbb{E}_{\nu_x^{\sigma,\beta,Id}}\left[ \left( Y - \mathbb{E}_{\nu_x^{\sigma,\beta,Id}}[Y] \right)^{\otimes 3} \right] = D_x^2 \left( \mathbb{E}_{\nu_x^{\sigma,1,Id}}[Y] \right)_{\beta x} = \alpha_3 x \otimes x \otimes x + \beta_3 Sym(x \otimes Id).
$$

Where $Sym(x \otimes Id) = x_i \delta_{jk} + x_j \delta_{ik} + x_k \delta_{ij}$. We need to compute the coefficients $\alpha_1, \alpha_2, \beta_2, \alpha_3, \beta_3$. Define $A(\beta) = \int_{S^{d-1}} \langle x, y \rangle \nu_x^{\sigma,\beta,Id}(dy)$. Similarly to what has been done in (8), one can relate this to the cumulants of $\langle x, Y \rangle$ by noticing that:

$$
\begin{aligned}
\partial_\beta^n (A(\beta))_\beta &= \partial_t^n (A(\beta+t))_{t=0} = \partial_t^n \left( \frac{\int e^{(\beta+t)\langle x,y \rangle} \langle x,y \rangle \sigma(dy)}{\int e^{(\beta+t)\langle x,y \rangle} \sigma(dy)} \right)_{t=0} \\
&= \partial_t^n \left( \frac{\int e^{t\langle x,y \rangle} \langle x,y \rangle e^{\beta \langle x,y \rangle} \sigma(dy)}{\int e^{t\langle x,y \rangle} e^{\beta \langle x,y \rangle} \sigma(dy)} \right)_{t=0} = \partial_t^n \left( \frac{\int e^{t\langle x,y \rangle} \langle x,y \rangle \nu_x^{\sigma,\beta,Id}(dy)}{\int e^{t\langle x,y \rangle} \nu_x^{\sigma,\beta,Id}(dy)} \right)_{t=0} \\
&= \partial_t^{n+1} \left( \log \mathbb{E}_{\nu_x^{\sigma,\beta,Id}}[e^{t\langle x,Y \rangle}] \right)_{t=0}.
\end{aligned}
$$

This give us immediately the following identities:

$$
\begin{aligned}
A(\beta) &= E_{\nu_x^{\sigma,\beta,Id}}[\langle x, Y \rangle], \\
A'(\beta) &= E_{\nu_x^{\sigma,\beta,Id}}[(\langle x, Y \rangle - A(\beta))^2], \\
A''(\beta) &= E_{\nu_x^{\sigma,\beta,Id}}[(\langle x, Y \rangle - A(\beta))^3].
\end{aligned}
\tag{11}
$$

Suppose without loss of generality that $x = e_1$. Then $\alpha_1$ is given by:

$$
\alpha_1 = \mathbb{E}_{\nu_x^{\sigma,\beta,Id}}[\langle e_1, Y \rangle] = A(\beta).
\tag{12}
$$

The coefficients $\alpha_2$ and $\beta_2$ can be obtained comparing the representations in (10) with the representations in (9), and exploiting the relations in (11):

$$
\begin{aligned}
\alpha_2 + \beta_2 &= D_x^1 \left( \mathbb{E}_{\nu_x^{\sigma,1,Id}}[Y] \right)_{\beta x} [e_1, e_1] = \mathbb{E}_{\nu_x^{\sigma,\beta,Id}}[(\langle e_1, Y \rangle - A(\beta))^2] = A'(\beta), \\
(d-1)\beta_2 &= \sum_{i>1}^{d} D_x^1 \left( \mathbb{E}_{\nu_x^{\sigma,1,Id}}[Y] \right)_{\beta x} [e_i, e_i] = \sum_{i>1}^{d} \mathbb{E}_{\nu_x^{\sigma,\beta,Id}}[(\langle e_i, Y \rangle)^2] \\
&= 1 - \mathbb{E}_{\nu_x^{\sigma,\beta,Id}}[(\langle e_1, Y \rangle)^2] = 1 - \mathbb{E}_{\nu_x^{\sigma,\beta,Id}}[(\langle e_1, Y \rangle - A)^2] - A^2 \\
&= 1 - A' - A^2.
\end{aligned}
\tag{13}
$$

And the same can be done for $\alpha_3$ and $\beta_3$:

$$
\begin{aligned}
\alpha_3 + 3\beta_3 &= D_x^2 \left( \mathbb{E}_{\nu_x^{\sigma,1,Id}}[Y] \right)_{\beta x} [e_1, e_1, e_1] = \mathbb{E}_{\nu_x^{\sigma,\beta,Id}}[(\langle e_1, Y \rangle - A(\beta))^3] = A''(\beta), \\
(d-1)\beta_3 &= \sum_{i>1}^{d} D_x^2 \left( \mathbb{E}_{\nu_x^{\sigma,1,Id}}[Y] \right)_{\beta x} [e_1, e_i, e_i] = \sum_{i>1}^{d} \mathbb{E}_{\nu_x^{\sigma,\beta,Id}}[(\langle e_i, Y \rangle)^2(\langle e_1, Y \rangle - A)] \\
&= -\mathbb{E}_{\nu_x^{\sigma,\beta,Id}}[(\langle e_1, Y \rangle)^2(\langle e_1, Y \rangle - A)] \\
&= -\mathbb{E}_{\nu_x^{\sigma,\beta,Id}}[(\langle e_1, Y \rangle - A)^3] - 2A\mathbb{E}_{\nu_x^{\sigma,\beta,Id}}[(\langle e_1, Y \rangle - A)^2] \\
&= -A'' - 2AA'.
\end{aligned}
$$

To conclude it is sufficient to show that $1 - A^2 = O(\frac{1}{\beta})$ and $A', A'' = O(\frac{1}{\beta^2})$. Now, using the identity:

$$
Z_\beta = \int_{S^{d-1}} e^{\beta \langle x, y \rangle} d\sigma(y) = C_d \beta^{1-d/2} I_{d/2-1}(\beta),
$$

we can explicitly compute $A(\beta)$ as:

$$
\begin{aligned}
A(\beta) &= \frac{\partial_\beta Z_\beta}{Z_\beta} = \frac{(1-\frac{d}{2})\beta^{-d/2} I_{d/2-1}(\beta) + \beta^{1-d/2}(I_{d/2}(\beta) + (\frac{d}{2}-1)\frac{1}{\beta} I_{d/2-1}(\beta))}{\beta^{1-d/2} I_{d/2-1}(\beta)} \\
&= \frac{I_{d/2}(\beta)}{I_{d/2-1}(\beta)} \approx 1 - \frac{d-1}{2\beta},
\end{aligned}
\tag{14}
$$

where we used the derivatives rules for the modified Bessel function:

$$
\begin{aligned}
I_\nu'(z) &= I_{\nu-1}(z) - \frac{\nu}{z} I_\nu(z), \\
I_\nu'(z) &= I_{\nu+1}(z) + \frac{\nu}{z} I_\nu(z),
\end{aligned}
$$

and its asymptotic behavior (in both cases see [1]).

In a similar way we can also compute:

$$
\begin{aligned}
A'(\beta) &= \frac{I_{d/2}(\beta)'}{I_{d/2-1}(\beta)} - \frac{I_{d/2}(\beta)}{\left(I_{d/2-1}(\beta)\right)^2} I_{d/2-1}(\beta)' \\
&= \frac{I_{d/2-1}(\beta) - \frac{d/2}{\beta} I_{d/2}(\beta)}{I_{d/2-1}(\beta)} - \frac{I_{d/2}(\beta)}{I_{d/2-1}(\beta)^2}\left(I_{d/2}(\beta) + \frac{d/2-1}{\beta} I_{d/2-1}(\beta)\right) \\
&= 1 - \frac{d/2}{\beta} A(\beta) - A(\beta)^2 - \frac{d/2-1}{\beta} A(\beta) \\
&= 1 - A^2(\beta) - \frac{d-1}{\beta} A(\beta) \approx \frac{(d-1)^2}{4\beta^2},
\end{aligned}
\tag{15}
$$

and

$$
A''(\beta) = -2A(\beta)A'(\beta) - \frac{d-1}{\beta} A'(\beta) \approx O(\frac{1}{\beta^2}).
$$

where we used the asymptotics in (14). This is sufficient to conclude the proof. $\square$

**Lemma A.7.** *Given a strictly positive probability measure $\mu$ the following holds:*

$$
|\nu_x^{\mu,\beta,B}(y) - \nu_x^{\sigma,\beta,B}(y)| \le \left(\frac{\|\nabla\mu\|_\infty}{\min|\mu|}(|y - x_B| + C\beta^{-1/2})\right)\nu_x^{\sigma,\beta,B},
$$

*where $x_B = \frac{B^T x}{|B^T x|}$*

*Proof.* Indeed:

$$
\begin{aligned}
\nu_x^{\mu,\beta,B}(y) &= \frac{e^{\beta\langle x, By\rangle}\mu(y)}{\int_{S^{d-1}} e^{\beta\langle x, By\rangle}\mu(y)} \\
&= \frac{e^{\beta\langle x, By\rangle}\mu(x_B) + e^{\beta\langle x, By\rangle}(\mu(y) - \mu(x_B))}{\mu(x_B)\int_{S^{d-1}} e^{\beta\langle x, By\rangle}d\sigma(y) + \int_{S^{d-1}} e^{\beta\langle x, By\rangle}(\mu(y) - \mu(x_B))d\sigma(y)} \\
&= \nu_x^{\sigma,\beta,B}(y)\left(\frac{1 + \frac{\mu(y) - \mu(x_B)}{\mu(x_B)}}{1 + \frac{1}{\mu(x_B)}\int(\mu(y) - \mu(x_B))d\nu_x^{\sigma,\beta,B}}\right) \\
&= \nu_x^{\sigma,\beta,B}(y)\left(1 + \frac{\frac{\mu(y) - \mu(x_B)}{\mu(x_B)} - \frac{1}{\mu(x_B)}\int(\mu(y) - \mu(x_B))d\nu_x^{\sigma,\beta,B}(y)}{1 + \frac{1}{\mu(x_B)}\int(\mu(y) - \mu(x_B))d\nu_x^{\sigma,\beta,B}(y)}\right) \\
&= \nu_x^{\sigma,\beta,B}(y)\left(1 + (R_1 + R_2)(1 + R_2)\right) \\
&\le \nu_x^{\sigma,\beta,B}(y)\left(1 + \frac{\|\nabla\mu\|_\infty}{\min|\mu|}(|y - x_B| + C\beta^{-1/2})\right),
\end{aligned}
$$

where we used:

$$
|R_1| \le \frac{|\mu(y) - \mu(x_B)|}{\mu(x_B)} \le \frac{\|\nabla\mu\|_\infty}{\min|\mu|}|y - x|,
$$

$$
|R_2| \le \frac{1}{\mu(x_B)}\int(\mu(y) - \mu(x_B))d\nu_x^{\sigma,\beta,B} \le \frac{\|\nabla\mu\|_\infty}{\min|\mu|}\int|y - x_B|d\nu_x^{\sigma,\beta,B} \le C\beta^{-1/2}\frac{\|\nabla\mu\|_\infty}{\min|\mu|},
$$

and the last inequality is a consequence of $\nu_x^{\sigma,\beta,B} = \nu_{x_B}^{\sigma,\beta|B^T x|,Id}$ and lemma D.1. $\square$

**Proposition A.8.** *The derivatives of the vector field $\chi^\beta[\mu]$ satisfy:*

$$
\chi[\mu] \le C,
$$

$$
D_x^1\chi[\mu] \le C\left(1 + \frac{\|\nabla\mu\|_\infty}{\min|\mu|}\beta^{-1/2}\right),
$$

$$
D_x^2\chi[\mu] \le C\left(1 + \frac{\|\nabla\mu\|_\infty}{\min|\mu|}\beta^{-1/2} + \frac{\|\nabla\mu\|_\infty}{\min|\mu|}\right).
$$

*Proof.* Thanks to Lemma A.5 we just need to bound the cumulants. The first one is already done.

Second cumulant:

$$\left| \mathbb{E}_{\nu_x^{\mu,\beta,B}}\left[ \left(Y - \mathbb{E}_{\nu_x^{\mu,\beta,B}}[Y]\right)^{\otimes 2} \right] \right| = \left| \int \int (y - z_1) \otimes (y - z_2) \nu_x^{\mu,\beta,B}(dy) \nu_x^{\mu,\beta,B}(dz_1) \nu_x^{\mu,\beta,B}(dz_2) \right|$$

$$= \left| \mathbb{E}_{\nu_x^{\sigma,\beta,B}}\left[ \left(Y - \mathbb{E}_{\nu_x^{\sigma,\beta,B}}[Y]\right)^{\otimes 2} \right] \right| + R.$$

We have already shown in lemma A.6 that the first term is $\leq C\frac{1}{\beta}$. Now we need to bound the second term. $R$ can be expanded by multi-linearity and using lemma A.7 the worst case is either of the form:

$$\leq C \frac{\|\nabla\mu\|_\infty}{\min|\mu|} \beta^{-1/2} \left| \int \int (y - z_1) \otimes (y - z_2) \nu_x^{\sigma,\beta,B}(dy) \nu_x^{\sigma,\beta,B}(dz_1) \nu_x^{\sigma,\beta,B}(dz_2) \right|$$

$$\leq C \frac{\|\nabla\mu\|_\infty}{\min|\mu|} \beta^{-1/2} \int \nu_x^{\sigma,\beta,B}(dy) \prod_{i=1}^2 \int |y - z_i| \nu_x^{\sigma,\beta,B}(dz_i)$$

$$= C \frac{\|\nabla\mu\|_\infty}{\min|\mu|} \beta^{-1/2} \int \nu_x^{\sigma,\beta,B}(dy) \left( \int |y - z_1| \nu_x^{\sigma,\beta,B}(dz_1) \right)^2$$

$$\leq C \frac{\|\nabla\mu\|_\infty}{\min|\mu|} \beta^{-1/2} \int \nu_x^{\sigma,\beta,B}(dy) \int |y - z_1|^2 \nu_x^{\sigma,\beta,B}(dz_1)$$

$$\leq 4C \frac{\|\nabla\mu\|_\infty}{\min|\mu|} \beta^{-1/2} \int |y - x_B|^2 \nu_x^{\sigma,\beta,B}(dy) \leq 4C \frac{\|\nabla\mu\|_\infty}{\min|\mu|} \beta^{-1/2}\beta^{-1} \leq C \frac{\|\nabla\mu\|_\infty}{\min|\mu|} \frac{1}{\beta^{3/2}},$$

where in the last line we used lemma D.1, or of the form:

$$\leq C \frac{\|\nabla\mu\|_\infty}{\min|\mu|} \left| \int \int (y - z_1) \otimes (y - z_2)|y - z_B| \nu_x^{\sigma,\beta,B}(dy) \nu_x^{\sigma,\beta,B}(dz_1) \nu_x^{\sigma,\beta,B}(dz_2) \right|$$

$$\leq C \frac{\|\nabla\mu\|_\infty}{\min|\mu|} \int \int |y - z_1||y - z_2||y - z_B| \nu_x^{\sigma,\beta,B}(dy) \nu_x^{\sigma,\beta,B}(dz_1) \nu_x^{\sigma,\beta,B}(dz_2)$$

$$\leq C \frac{\|\nabla\mu\|_\infty}{\min|\mu|} \left( \int |y - z_B|^2 \nu_x^{\sigma,\beta,B}(dy) \right)^{1/2} \left( \int \int |y - z|^4 \nu_x^{\sigma,\beta,B}(dy) \nu_x^{\sigma,\beta,B}(dz) \right)^{1/2}$$

$$\leq 4C \frac{\|\nabla\mu\|_\infty}{\min|\mu|} \left( \int |y - z_B|^2 \nu_x^{\sigma,\beta,B}(dy) \right)^{1/2} \left( \int |y - z_B|^4 \nu_x^{\sigma,\beta,B}(dy) + \int |z - z_B|^4 \nu_x^{\sigma,\beta,B}(dz) \right)^{1/2}$$

$$\leq C \frac{\|\nabla\mu\|_\infty}{\min|\mu|} \beta^{-1/2}\beta^{-1} \leq C \frac{\|\nabla\mu\|_\infty}{\min|\mu|} \frac{1}{\beta^{3/2}},$$

or of the form:

$$\leq C \frac{\|\nabla\mu\|_\infty}{\min|\mu|} \left| \int \int (y - z_1) \otimes (y - z_2)|z_1 - z_B| \nu_x^{\sigma,\beta,B}(dy) \nu_x^{\sigma,\beta,B}(dz_1) \nu_x^{\sigma,\beta,B}(dz_2) \right|$$

$$\leq C \frac{\|\nabla\mu\|_\infty}{\min|\mu|} \int \int |y - z_1||y - z_2||z_1 - z_B| \nu_x^{\sigma,\beta,B}(dy) \nu_x^{\sigma,\beta,B}(dz_1) \nu_x^{\sigma,\beta,B}(dz_2)$$

$$\leq C \frac{\|\nabla\mu\|_\infty}{\min|\mu|} \left( \int |z_1 - z_B|^2 \nu_x^{\sigma,\beta,B}(dz_1) \right)^{1/2} \left( \int \int |y - z_1|^2 |y - z_2|^2 \nu_x^{\sigma,\beta,B}(dy) \nu_x^{\sigma,\beta,B}(dz_1) \nu_x^{\sigma,\beta,B}(dz_2) \right)^{1/2}$$

$$\leq 4C \frac{\|\nabla\mu\|_\infty}{\min|\mu|} \left( \int |y - z_B|^2 \nu_x^{\sigma,\beta,B}(dy) \right)^{1/2} \left( \int |y - z_B|^4 \nu_x^{\sigma,\beta,B}(dy) + \int |z - z_B|^4 \nu_x^{\sigma,\beta,B}(dz) \right)^{1/2}$$

$$\leq C \frac{\|\nabla\mu\|_\infty}{\min|\mu|} \beta^{-1/2}\beta^{-1} \leq C \frac{\|\nabla\mu\|_\infty}{\min|\mu|} \frac{1}{\beta^{3/2}}.$$

These are the worst cases because every $|y - z|$ produces an additional $\beta^{-1/2}$ by lemma D.1. Hence we proved, thanks to lemma A.5, that:

$$|D_x^1 \chi[\mu]| \leq C \left( 1 + \frac{\|\nabla\mu_\beta\|_\infty}{\min|\mu|} \beta^{-1/2} \right).$$

The bound for the third cumulant is similar to what we have done above:

$$|\mathbb{E}_{\nu_x^{\mu,\beta,B}}[(Y - \mathbb{E}_{\nu_x^{\mu,\beta,B}}[Y])^{\otimes 3}]|$$

$$= \left| \int \int \int \int (y-z_1) \otimes (y-z_2) \otimes (y-z_3) \nu_x^{\mu,\beta,B}(dy) \nu_x^{\mu,\beta,B}(dz_1) \nu_x^{\mu,\beta,B}(dz_2) \nu_x^{\mu,\beta,B}(dz_3) \right|$$

$$= |\mathbb{E}_{\nu_x^{\sigma,\beta,B}}[(Y - \mathbb{E}_{\nu_x^{\sigma,\beta,B}}[Y])^{\otimes 3}]| + R.$$

We have already shown that the first term is $O(\frac{1}{\beta^2})$. Now we need to bound the second term. $R$ can be expanded again as in lemma A.7 and the worst case is either of the form:

$$\leq C\frac{\|\nabla\mu\|_\infty}{\min|\mu|}\beta^{-1/2}\left| \int \int \int \int (y-z_1) \otimes (y-z_2) \otimes (y-z_3) \nu_x^{\sigma,\beta,B}(dy) \nu_x^{\sigma,\beta,B}(dz_1) \nu_x^{\sigma,\beta,B}(dz_2) \nu_x^{\sigma,\beta,B}(dz_3) \right|$$

$$\leq C\frac{\|\nabla\mu\|_\infty}{\min|\mu|}\beta^{-1/2} \int \nu_x^\sigma(dy) \prod_{i=1}^{3} \int |y - z_i| \nu_x^{\sigma,\beta,B}(dz_i)$$

$$= C\frac{\|\nabla\mu\|_\infty}{\min|\mu|}\beta^{-1/2} \int \nu_x^\sigma(dy) \left( \int |y - z_1| \nu_x^{\sigma,\beta,B}(dz_1) \right)^3$$

$$\leq C\frac{\|\nabla\mu\|_\infty}{\min|\mu|}\beta^{-1/2} \int \nu_x^\sigma(dy) \int |y - z_1|^3 \nu_x^{\sigma,\beta,B}(dz_1)$$

$$\leq 2C\frac{\|\nabla\mu\|_\infty}{\min|\mu|}\beta^{-1/2} \int |y - x_B|^3 \nu_x^{\sigma,\beta,B}(dy) \leq C\beta^{-1/2}\beta^{-3/2} \leq C\frac{\|\nabla\mu\|_\infty}{\min|\mu|}\frac{1}{\beta^2},$$

or of the form:

$$\leq C\frac{\|\nabla\mu\|_\infty}{\min|\mu|} \left| \int \int \int \int (y-z_1) \otimes (y-z_2) \otimes (y-z_3)|y - z_B| \nu_x^{\sigma,\beta,B}(dy) \nu_x^{\sigma,\beta,B}(dz_1) \nu_x^{\sigma,\beta,B}(dz_2) \nu_x^{\sigma,\beta,B}(dz_3) \right|$$

$$\leq C\frac{\|\nabla\mu\|_\infty}{\min|\mu|} \int \int \int \int |y - z_1||y - z_2||y - z_3||y - z_B| \nu_x^{\sigma,\beta,B}(dy) \nu_x^{\sigma,\beta,B}(dz_1) \nu_x^{\sigma,\beta,B}(dz_2) \nu_x^{\sigma,\beta,B}(dz_3)$$

$$\leq C\frac{\|\nabla\mu\|_\infty}{\min|\mu|} \prod_{i=1}^{3} \left( \int \int |y - z_i|^4 \nu_x^{\sigma,\beta,B}(dz_i) \nu_x^{\sigma,\beta,B}(dy) \right)^{1/4} \left( \int |y - z_B|^4 \nu_x^{\sigma,\beta,B}(dy) \right)^{1/4}$$

$$= C\frac{\|\nabla\mu\|_\infty}{\min|\mu|} \left( \int \int |y - z|^4 \nu_x^{\sigma,\beta,B}(dz) \nu_x^{\sigma,\beta,B}(dy) \right)^{3/4} \left( \int |y - z_B|^4 \nu_x^{\sigma,\beta,B}(y) \right)^{1/4}$$

$$\leq C\frac{\|\nabla\mu\|_\infty}{\min|\mu|} (\beta^{-2})^{3/4}(\beta^{-2})^{1/4} = C\frac{\|\nabla\mu\|_\infty}{\min|\mu|}\beta^{-2}.$$

Thus, we can replace the bounds on the second and third cumulants that we obtained above in the estimates of Lemma A.5 to conclude that:

$$|D_x^2\chi[\mu]| \leq C \left( 1 + \beta |\mathbb{E}_{\nu_x^{\mu,\beta,B}}[(Y - \mathbb{E}_{\nu_x^{\mu,\beta,B}}[Y])^{\otimes 2}]| + \beta^2 |\mathbb{E}_{\nu_x^{\mu,\beta,B}}[(Y - \mathbb{E}_{\nu_x^{\mu,\beta,B}}[Y])^{\otimes 3}]| \right)$$

$$\leq C \left( 1 + \frac{\|\nabla\mu_\beta\|_\infty}{\min|\mu|}\beta^{-1/2} + \frac{\|\nabla\mu_\beta\|_\infty}{\min|\mu|} \right).$$

$\square$

**Lemma A.9.** *If $\mu$ solves the PDE:*

$$\begin{cases} \partial_t\mu &= -div(\mu\chi[\mu]), \\ \mu(0) &= \mu_0. \end{cases}$$

*then:*

- $\partial_t\|\mu\|_\infty \leq \|\mu\|_\infty|D_x^1\chi[\mu]|,$

- $\partial_t \left( \min \mu^\beta \right) \geq - \left( \min \mu^\beta \right) |D_x^1 \chi[\mu]|$,

- $\partial_t \|\nabla\mu\|_\infty \leq \|\nabla\mu\|_\infty |D_x^1 \chi[\mu]| + \frac{1}{2}\|\mu\|_\infty |D_x^2 \chi[\mu]|$.

*Proof.* Let $x_t$ be a point of maximum for $|\mu_t|$. Then $\nabla_{S^{d-1}} \mu_t(x_t) = 0$ and:

$$\partial_t \mu_t(x_t) = -div(\mu\chi[\mu])(x_t) + \langle \nabla\mu_t(x_t), x_t' \rangle = -\mu_t(x_t) div(\chi[\mu])(x_t).$$

And for $\min \mu$ we can use the same argument.

Now, let $x_t$ be a point of maximum for $|\nabla_{S^{d-1}}\mu|^2$, then $H_{S^{d-1}}\mu(x_t)\nabla_{S^{d-1}}\mu(x_t) = 0$, hence:

$$\begin{aligned}
\partial_t |\nabla\mu(x_t)|^2 &= - \langle \nabla\mu(x_t), \nabla div(\mu\chi[\mu])(x_t) \rangle + \langle \nabla\mu(x_t), H\mu(x_t)x_t' \rangle \\
&= - \langle \nabla\mu(x_t), \nabla(\nabla\mu \cdot \chi[\mu])(x_t) \rangle - \langle \nabla\mu(x_t), \nabla(\mu D_x^1\chi[\mu]))(x_t) \rangle \\
&= - \langle \nabla\mu(x_t), H\mu(x_t)\chi[\mu](x_t) \rangle - \langle \nabla\mu(x_t), D_x^1\chi[\mu](x_t)\nabla\mu(x_t) \rangle \\
&\quad - \langle \nabla\mu(x_t), D_x^1\chi[\mu](x_t)\nabla\mu(x_t) \rangle - \langle \nabla\mu(x_t), D_x^2\chi[\mu](x_t)\mu(x_t) \rangle \\
&\leq 2|\nabla\mu(x_t)|^2|D_x^1\chi[\mu]| + |\nabla\mu(x_t)||\mu(x_t)||D_x^2\chi[\mu]|.
\end{aligned}$$

Using that $\partial_t |\nabla\mu(x_t)|^2 = 2|\nabla\mu(x_t)|\partial_t|\nabla\mu(x_t)|$ and dividing by $|\nabla\mu(x_t)|$ we get the thesis. $\qquad\square$

**Lemma A.10.** *Consider again $\mu_t$ solution of the PDE:*

$$\begin{cases} \partial_t \mu &= -div(\mu\chi_\beta[\mu]), \\ \mu(0) &= \mu_0. \end{cases}$$

*Define:*

$$\begin{aligned}
C_1 &= 2C\|\mu_0\|_\infty e^{2CT}, \\
C_2 &= 2C\left(1 + \frac{\|\mu_0\|_\infty}{\min\mu_0}e^{4CT}\right),
\end{aligned}$$

*Then, for $\beta$ large enough (depending just on $\mu_0$ and $C$):*

- $\|\mu_t\|_\infty \leq 2(\|\mu_0\|_\infty)e^{2Ct}$,

- $\min\mu_t \geq \frac{1}{2}(\min\mu_0)e^{-2Ct}$,

- $|\nabla\mu_t|_\infty \leq 2\left(\frac{C_1}{C_2} + |\nabla\mu_0|_\infty\right)e^{C_2 t}$.

*Proof.* The thesis is true at time $t = 0$. Let us assume that it is true on $[0, t]$. Then $\exists\beta$ big enough (where "big" depends only on $C_1, C_2$, i.e. just $\mu_0, B, V, d$) such that $\frac{\|\nabla\mu\|_\infty}{\min|\mu|}\beta^{-1/2} \leq 1$ on $[0, t]$. Hence $D\chi[\mu_\beta] \leq 2C$ on $[0, t]$ thanks to proposition A.8. By Gronwall applied to the first two bounds in lemma A.9 we can conclude:

$$\begin{aligned}
\|\mu_t\|_\infty &\leq (\|\mu_0\|_\infty)e^{2Ct}, \\
\min\mu_t &\geq (\min\mu_0)e^{-2Ct},
\end{aligned}$$

For $\|\nabla\mu\|_\infty$ we have, again by lemma A.9:

$$\begin{aligned}
\partial_t\|\nabla\mu_t\|_\infty &\leq \|\nabla\mu_t\|_\infty |D_x^1\chi[\mu_t]| + \frac{1}{2}\|\mu\|_\infty |D_x^2\chi[\mu_t]| \\
&\leq 2C\|\nabla\mu_t\|_\infty + \left(\|\mu_0\|_\infty e^{2CT}\right)C\left(1 + 1 + \frac{\|\nabla\mu_t\|_\infty}{\min|\mu|}\right) \\
&\leq 2C\|\mu_0\|_\infty e^{2CT} + C\left(2 + 2\frac{\|\mu_0\|_\infty}{\min\mu_0}e^{4CT}\right)\|\nabla\mu_t\|_\infty \\
&= C_1 + C_2\|\nabla\mu_t\|_\infty.
\end{aligned}$$

where in the second row we used the assumption on $[0, t]$ and proposition A.8. Hence by Gronwall:

$$\|\nabla \mu_t\|_\infty \leq \left(\frac{C_1}{C_2} + \|\nabla \mu_0\|_\infty\right) e^{C_2 t}.$$

This concludes the continuation argument and the proof. □

**Corollary A.11.** *For $\beta$ large enough (depending on $\mu_0, B, V, d$) the vector fields $\{\chi_\beta[\mu]\}_{\beta,t}$ are jointly Lipschitz in $\beta$ and $t \in [0, T]$.*

*Proof.* This is a consequence of lemma A.10 and proposition A.8. □

**Corollary A.12.** *For every $x \in \mathbb{S}^{d-1}$ and $t \in [0, T]$:*

$$\chi_\beta[\mu_t^\beta](x) \to \frac{P_x V B^T x}{|B^T x|} \quad as \quad \beta \to \infty$$

*Proof.* With the usual notations one can write:

$$\chi_\beta[\mu^\beta](x) = P_x V \left(\mathbb{E}_{\nu_x^{\mu,\beta,B}}[Y]\right) = P_x V \left(\mathbb{E}_{\nu_x^{\sigma,\beta,B}}[Y]\right) + R.$$

The reminder $R$ is bounded using lemma A.7 by:

$$|R| \leq C \left(\beta^{-1/2} + \frac{\|\nabla \mu_t^\beta\|_\infty}{\min \mu_t^\beta} \int |y - x_B| \nu_x^{\sigma,\beta,B}\right)$$

$$\leq C \left(\beta^{-1/2} + \frac{\|\nabla \mu_t^\beta\|_\infty}{\min \mu_t^\beta} \beta^{-1/2}\right) = O(\beta^{-1/2}),$$

where the last line follows from lemma D.1 and lemma A.10. Hence, the proof can be concluded by noticing that:

$$P_x V \left(\mathbb{E}_{\nu_x^{\sigma,\beta,B}}[Y]\right) = P_x V \left(\mathbb{E}_{\nu_{x_B}^{\sigma,\beta|B^T x|, Id}}[Y]\right) = P_x V(A(\beta) x_B) = (1 + O(\beta^{-1})) P_x \frac{V B^T x}{|B^T x|},$$

where we used the identities in equations 10, 12, 14. □

We can finally pass to the limit in the PDE. Indeed consider a subsequence $\mu_\beta$ of solutions to the PDE converging in $\mathcal{C}([0, T], \mathcal{P}(\mathbb{S}^{d-1}))$ to a certain probability measure $\mu^\infty$. If we define the vector field $\chi_\infty(x) := P_x V \frac{B^T x}{|B^T x|}$, then for every $f \in C_b^2(\mathbb{S}^{d-1})$:

$$\langle f, \mu_t^\infty\rangle - \langle f, \mu_0^\infty\rangle - \int_0^t \langle \nabla f, \chi_\infty \mu_s^\infty\rangle \, ds| \leq \langle f, \mu_t^\infty - \mu_t^\beta\rangle| + \int_0^t |\langle \nabla f, \chi_\infty \mu_s^\infty - \chi_\beta[\mu_s^\beta]\mu_s^\beta\rangle| \, ds,$$

where we used that $\mu_0^\infty = \mu_0 = \mu_0^\beta$ and that the PDE in weak form for $\mu^\beta$ is:

$$\langle f, \mu_t^\beta\rangle - \langle f, \mu_0^\beta\rangle - \int_0^t \langle \nabla f, \chi^\beta[\mu_s^\beta]\mu_s^\beta\rangle \, ds = 0.$$

Moreover, as $\beta \to \infty$:

$$|\langle f, \mu_t^\infty - \mu_t^\beta\rangle| \to 0$$

thanks to the fact that $f$ is Lipschitz and by the definition of convergence in $\mathcal{C}([0, T], \mathcal{P}(\mathbb{S}^{d-1}))$. For the second term:

$$\int_0^t |\langle \nabla f, \chi_\infty \mu_s^\infty - \chi_\beta[\mu_s^\beta]\mu_s^\beta\rangle| \, ds \leq \int_0^t |\langle \nabla f, (\chi_\infty - \chi_\beta[\mu_s^\beta])\mu_s^\infty\rangle| \, ds$$

$$+ \int_0^t |\langle \nabla f, \chi_\beta[\mu_s^\beta](\mu_s^\infty - \mu_s^\beta)\rangle| \, ds.$$

The first part goes to 0 by dominated convergence ($\nabla f, \chi_\infty, \chi^\beta[\mu_\beta]$ are bounded, and $\chi_\beta[\mu_\beta] \to \chi_\infty$ point-wise by lemma A.12). The second part goes to 0 by definition of the convergence $\mu_\infty \to \mu_\beta$ and by equi-lipschitzianity of $\chi_\beta[\mu_\beta]$ (see corollary A.11).

The uniqueness is standard, since $\mu$ is a probability measure and the vector field is smooth (see, for example, [5]).

## A.2 Asymptotic behavior

This section studies the asymptotic behavior of the support of the solution to the partial differential equation:

$$\partial_t \mu = -div \left( \mu \frac{P_x V B^T x}{|B^T x|} \right) \tag{16}$$

and in particular, we prove Proposition 3.4.

**Lemma A.13.** *The ODE:*

$$\frac{d}{dt} x(t) = \frac{P_x V B^T x(t)}{|B^T x(t)|} \tag{17}$$

*is a time-reparameterization of:*

$$\frac{d}{dt} y(t) = P_y V B^T y(t)$$

*where $y(t) = x(f^{-1}(t))$ and $f(t) = \int_0^t \frac{1}{|B^T x(s)|} ds$.*

**Remark A.14.** *The reparameterization is well defined since $B$ is invertible.*

*Proof.* : We have:

$$\frac{d}{dt} y(t) = x'(f^{-1}(t)) \cdot \frac{d}{dt} f^{-1}(t)$$
$$= \frac{P_x(V B^T x)}{|B^T x|} (f^{-1}(t)) \cdot |B^T x(f^{-1}(t))| = P_y(V B^T y)$$

This shows that the ODE is a time-reparameterization of the ODE for $y(t)$. $\square$

**Lemma A.15.** *If $z(t)$ solves:*

$$\frac{d}{dt} z(t) = V B^T z(t),$$

*then $y(t) = \frac{z(t)}{|z(t)|}$.*

*Proof.* We have:

$$\frac{d}{dt} y(t) = \frac{d}{dt} \left( \frac{z(t)}{|z(t)|} \right) = \frac{z'}{|z|} - \frac{1}{|z|^2} \frac{1}{2|z|} 2\langle z, z' \rangle z$$
$$= \frac{V B^T z}{|z|} - \frac{\langle z, V B^T z \rangle}{|z|^3} z = V B^T y - \langle y, V B^T y \rangle y$$
$$= P_y V B^T y.$$

This concludes the proof. $\square$

**Corollary A.16.** *For Lebesgue almost every $x_0$, the $\omega$-limit set $\omega(x_0) \subset E_{max}$.*

*Proof.* This is a consequence of lemma A.13, lemma A.15 and of the classical theory for linear ODEs, after reducing to the Jordan canonical form of the matrix $V B^T$. $\square$

**Remark A.17.** *This technical result parallels Lemma 3.1 in [29], which was used to analyze the dynamics of the first token in causal attention.*

And now we can finally prove Proposition 3.4:

*Proof.* Denote $\Phi_t$ the flow of the ODE (17) and let $\phi \in C_b^2(\mathbb{S}^{d-1})$ be a test function with $supp(\phi) \subset E_{max}^C \cap \mathbb{S}^{d-1}$. Fix $\mu_\infty \in \omega(\mu_0)$. Then there exists a divergent sequence of times $\{t_k\}_k$ such that $\mu_{t_k} \to \mu_\infty$ weakly. As a consequence:

$$\int_{\mathbb{S}^{d-1}} \phi(x)\mu_\infty(dx) = \lim_{k\to\infty} \int_{\mathbb{S}^{d-1}} \phi(x)\mu_{t_k}(dx) = \lim_{k\to\infty} \int_{\mathbb{S}^{d-1}} \phi(x)\Phi_{t_k\#}\mu_0(dx)$$

$$= \lim_{k\to\infty} \int_{\mathbb{S}^{d-1}} \phi(\Phi_{t_k}(x))\mu_0(dx) = 0,$$

where we used corollary A.16 and the dominated convergence theorem. $\qquad\square$

# B    Proofs of the heat phase

In this section, we prove Proposition 3.9, which characterizes the second phase using the heat equation on the sphere.

**Lemma B.1.** *Given a measure $\mu \in C^2(\mathbb{S}^{d-1}) \cap \mathcal{P}(\mathbb{S}^{d-1})$ strictly positive, then the following holds:*

$$\beta\chi_\beta[\mu](x) = \frac{\nabla_x\mu(x)}{\mu(x)} + \frac{\|\nabla\mu\|_\infty}{\min\mu}\left(1 + \frac{\|H_x\mu\|_\infty}{\min\mu}\right)O_{L^\infty(\mathbb{S}^{d-1})}(\beta^{-1/2}),$$

*with the gradient $\nabla_x$ and Hessian $H_x$ defined with respect to the standard Riemannian metric on $\mathbb{S}^{d-1}$.*

*Proof.*

$$\beta\chi_\beta[\mu](x) = \beta\frac{\int e^{\beta\langle x,y\rangle}P_x y\,\mu(dy)}{\int e^{\beta\langle x,y\rangle}\mu(dy)}$$

$$= \beta\frac{\int e^{\beta\langle x,y\rangle}P_x y\,\mu(dy)}{\mu(x)\int e^{\beta\langle x,y\rangle}\sigma(dy)}\frac{1}{1+R_1}$$

$$= \beta\left(\frac{\int e^{\beta\langle x,y\rangle}P_x y\,(\mu(x) + \langle y - x, \nabla\mu(x)\rangle)\,\sigma(dy)}{\mu(x)\int e^{\beta\langle x,y\rangle}\sigma(dy)} + R_2\right)\frac{1}{1+R_1},$$

where, by lemma D.1:

$$|R_1| := \left|\frac{\int e^{\beta\langle x,y\rangle}\mu(dy)}{\mu(x)\int e^{\beta\langle x,y\rangle}\sigma(dy)} - 1\right| \le \frac{\|\nabla\mu\|_\infty}{\min\mu}\int|y - x|\nu_x^{\sigma,\beta,I}(dy) = \frac{\|\nabla\mu\|_\infty}{\min\mu}O(\beta^{-1/2}),$$

$$|R_2| := \left|\frac{\int e^{\beta\langle x,y\rangle}P_x y\mu(dy) - \int e^{\beta\langle x,y\rangle}P_x y\,(\mu(x) + \langle y - x, \nabla\mu(x)\rangle)\,\sigma(dy)}{\mu(x)\int e^{\beta\langle x,y\rangle}\sigma(dy)}\right| \tag{18}$$

$$\le \frac{\|H\mu\|_\infty}{\min\mu}\int|y - x|^3\nu_x^{\sigma,\beta,I}(dy) = \frac{\|H\mu\|_\infty}{\min\mu}O(\beta^{-3/2}).$$

Hence:

$$\beta\chi_\beta[\mu](x) = \beta\left(P_x\int y\nu_x^{\sigma,\beta,I}(dy) + P_x\int(y - x)^{\otimes 2}\nu_x^{\sigma,\beta,I}(dy)\frac{\nabla_x\mu(x)}{\mu(x)} + R_2\right)\frac{1}{1+R_1},$$

and noticing that $\mathbb{E}_{\nu_x^{\sigma,\beta,I}}[Y]$ is parallel to $x$ (see proof of lemma A.6):

$$\beta\chi_\beta[\mu](x) = \beta\left(P_x\int(y - \mathbb{E}_{\nu_x^{\sigma,\beta,I}}[Y])^{\otimes 2}\nu_x^{\sigma,\beta,I}(dy)\frac{\nabla_x\mu(x)}{\mu(x)} + R_2\right)\frac{1}{1+R_1}$$

$$= \beta\left(P_x\mathbb{E}_{\nu_x^{\sigma,\beta,I}}[(Y - \mathbb{E}_{\nu_x^{\sigma,\beta,I}}[Y])^{\otimes 2}]\frac{\nabla_x\mu(x)}{\mu(x)} + R_2\right)\frac{1}{1+R_1}$$

$$= \beta\left(P_x(\alpha_2 x \otimes x + \beta_2 I)\frac{\nabla_x\mu(x)}{\mu(x)} + R_2\right)\frac{1}{1+R_1}$$

$$= \beta\left(\frac{1 - A'(\beta) - A(\beta)^2}{d - 1}\frac{\nabla_x\mu(x)}{\mu(x)} + R_2\right)\frac{1}{1+R_1},$$

where $\alpha_2, \beta_2$ are defined in the proof of Lemma A.6) and we used equation 13. To conclude, it suffices to replace equations 18 and the asymptotic estimates 14 and 15:

$$= \beta \left( \frac{1}{\beta} \frac{\nabla_x \mu(x)}{\mu(x)} + \frac{\|\nabla \mu\|_\infty}{\min \mu} O(\beta^{-3/2}) \right) \frac{1}{1 + \frac{\|H_x \mu\|_\infty}{\min \mu} O(\beta^{-1/2})}$$

$$= \left( \frac{\nabla_x \mu(x)}{\mu(x)} + \frac{\|\nabla \mu\|_\infty}{\min \mu} O(\beta^{-1/2}) \right) \left( 1 + \frac{\|H_x \mu\|_\infty}{\min \mu} O(\beta^{-1/2}) \right)$$

$$= \frac{\nabla_x \mu(x)}{\mu(x)} + \frac{\|\nabla \mu\|_\infty}{\min \mu} \left( 1 + \frac{\|H_x \mu\|_\infty}{\min \mu} \right) O(\beta^{-1/2}).$$

$\square$

**Corollary B.2.** *Given a family $\{\mu^\beta\}_\beta$ of probability measures on $\mathbb{S}^{d-1}$, suppose that there exist $c, C > 0$ such that $\|\mu^\beta\|_{C^2(\mathbb{S}^{d-1})} \leq C$ and $\mu^\beta \geq c$ for every $\beta \geq 0$. Moreover, assume there exists $\mu^\infty$ such that $\mu^\beta \to \mu^\infty$ in $C^1(\mathbb{S}^{d-1})$. Then*

$$\beta \chi_\beta[\mu^\beta](x) \to \frac{\nabla_x \mu^\infty(x)}{\mu^\infty(x)} \quad \forall x \in \mathbb{S}^{d-1},$$

*where $\nabla_x$ is the gradient with respect to the standard Riemannian metric on $\mathbb{S}^{d-1}$.*

*Proof of Proposition 3.9.* Without loss of generality, set $\gamma = -1$, the other case is analogous. The residual term is given by:

$$R_\beta = \beta \operatorname{div}(\mu \chi_\beta[\mu]) - \Delta \mu = \operatorname{div} \left( \mu \left[ \beta \chi_\beta[\mu] - \frac{\nabla \mu}{\mu} \right] \right).$$

It is sufficient to show that:

$$\left\| \beta \chi_\beta[\mu] - \frac{\nabla \mu}{\mu} \right\|_{C^{k+1}(\mathbb{S}^{d-1})} \to 0 \quad \text{as } \beta \to \infty.$$

Corollary B.2 guarantees convergence in $C^0(\mathbb{S}^{d-1})$ thanks to the assumptions on $\mu_t$. To improve this to higher regularity, we can use an interpolation argument through uniform bounds in $C^{k+2}$.

Define the kernel $W_\beta(t) := \frac{e^{\beta t}}{K_\beta}$, where $K_\beta := \int_{\mathbb{S}^{d-1}} e^{\beta \langle x, y \rangle} \, d\sigma(y)$. Then,

$$\chi_\beta[\mu](x) = \frac{\nabla(W_\beta * \mu)(x)}{(W_\beta * \mu)(x)}.$$

By the product rule, for every $0 \leq j \leq k+2$, there exists a polynomial $p_j$ such that:

$$\left\| D_x^j \left( \frac{\nabla(W_\beta * \mu)}{W_\beta * \mu} - \frac{\nabla \mu}{\mu} \right) \right\| \leq p_j \left( \|W_\beta * \mu\|_{C^{j+1}}, \|\mu\|_{C^{j+1}}, \min_{x \in \mathbb{S}^{d-1}} \mu \right),$$

where we used that $\min_{x \in \mathbb{S}^{d-1}} W_\beta * \mu \geq \min_{x \in \mathbb{S}^{d-1}} \mu$. The only thing left is to notice that

$$\|W_\beta * \mu\|_{C^j} \leq C_k \|\mu\|_{C^j},$$

though proving this on $\mathbb{S}^{d-1}$ requires some care.

Consider the case $j = 1$ ($j > 1$ follows by induction) and fix $v \in T_x(\mathbb{S}^{d-1})$. Let $A \in \mathfrak{so}(d)$ (a skew-symmetric matrix) satisfying $Ax = v$, and define $R(t) := e^{tA}$. In such a way $R(0)x = x$ and $R'(0)x = v$. Then:

$$\nabla_x(W_\beta * \mu)[v] = \frac{d}{dt}(W_\beta * \mu)(R(t)x)\Big|_{t=0}$$

$$= \frac{d}{dt} \int_{\mathbb{S}^{d-1}} W_\beta(\langle R(t)x, y \rangle) \mu(y) \, d\sigma(y)\Big|_{t=0}$$

$$= \frac{d}{dt} \int_{\mathbb{S}^{d-1}} W_\beta(\langle x, R(t)^T y \rangle) \mu(y) d\sigma(y)|_{t=0}$$

$$= \frac{d}{dt} \int_{\mathbb{S}^{d-1}} W_\beta(\langle x, z \rangle) \mu(R(t)z) d\sigma(y)|_{t=0}$$

$$= \int_{\mathbb{S}^{d-1}} W_\beta(\langle x, z \rangle) \nabla_z \mu[Az] \, d\sigma(z),$$

where we used the change of variable $z = R(t)^T y$, and the invariance of the measure on the sphere. Since $\|W_\beta\|_{L^1} = 1$, it follows that

$$\|\nabla_x (W_\beta * \mu)\|_{C^1(\mathbb{S}^{d-1})} \leq C\|\mu\|_{C^1(\mathbb{S}^{d-1})}.$$

Higher derivatives follow similarly, completing the proof. $\qquad\square$

## C   Proofs of the pairing phase

In this section we provide the proof of Proposition 3.11. Consider the interacting particle system on $\mathbb{S}^{d-1}$ described by the following ODEs corresponding to the case $(Q^T K = V = Id)$:

$$\dot{x}_i(t) = \frac{1}{Z_\beta(x_i)} \sum_{j=1}^N e^{\beta \langle x_i, x_j \rangle} P_{x_i}(x_j).$$

where $Z_\beta(x_i) = \sum_{j=1}^N e^{\beta \langle x_i, x_j \rangle}$ and $P_{x_i}(x_j) = x_j - \langle x_i, x_j \rangle x_i$ is the projection on the hyperplane orthogonal to $x_i$. Suppose that there exists a unique pair $(\underline{i}, \underline{j})$ such that at initialization $\langle x_{\underline{i}}, x_{\underline{j}} \rangle = \max_{i \neq j} \langle x_i, x_j \rangle$ and denote $\langle x_{\underline{i}}(t), x_{\underline{j}}(t) \rangle := d_t$. Define also $m_t := \max\{\langle x_i, x_j \rangle | i \neq j$ and $\{i,j\} \neq \{\underline{i}, \underline{j}\}\}$.

Let $\alpha := \arccos(m_0) - \arccos(d_0) > 0$ and consider the time rescaling given by the inverse of $d\tau = e^{\beta(1-d_t)}dt$, that we will still denote by $t$. Then:

$$\dot{x}_i(t) = \frac{e^{\beta(1-d_t)}}{Z_\beta(x_i)} \sum_{j=1}^N e^{\beta \langle x_i, x_j \rangle} P_{x_i}(x_j).$$

As usual the constant $C$ can change from line to line, but it does not depend on $\alpha$ or $\beta$.

**Lemma C.1.** *If $\beta$ is such that $Ce^{-\beta(1-\cos(\alpha/4))}T \leq \frac{1}{2}\alpha$, then:*

$$d_t - m_t \geq 1 - \cos\left(\frac{\alpha}{4}\right) \text{ on } [0, T].$$

*Proof.* We proceed by a standard continuation argument. At $t = 0$ we have $d_0 - m_0 \geq 1 - \cos(\alpha) \geq 1 - \cos(\alpha/4)$. Suppose the thesis holds on $[0, t]$. Then we have:

- if $i \neq \underline{i}$ and $j \neq \underline{j}$:

$$
\begin{aligned}
\partial_t \arccos(\langle x_i, x_j \rangle) &= -\frac{1}{\sqrt{1 - \langle x_i(t), x_j(t) \rangle^2}} \partial_t \langle x_i(t), x_j(t) \rangle \\
&= -\frac{1}{\sqrt{1 - \langle x_i, x_j \rangle^2}} \frac{e^{\beta(1-d_t)}}{Z_\beta(x_i)} \sum_{k=1}^N e^{\beta \langle x_i, x_k \rangle} \langle P_{x_i}(x_k), x_j \rangle \\
&\quad - \frac{1}{\sqrt{1 - \langle x_i, x_j \rangle^2}} \frac{e^{\beta(1-d_t)}}{Z_\beta(x_j)} \sum_{k=1}^N e^{\beta \langle x_j, x_k \rangle} \langle P_{x_j}(x_k), x_i \rangle \\
&= -\frac{e^{\beta(1-d_t)}}{Z_\beta(x_i)} \sum_{k=1}^N e^{\beta \langle x_i, x_k \rangle} \langle P_{x_i}(x_k), \frac{P_{x_i} x_j}{|P_{x_i} x_j|} \rangle \\
&\quad - \frac{e^{\beta(1-d_t)}}{Z_\beta(x_j)} \sum_{k=1}^N e^{\beta \langle x_j, x_k \rangle} \langle P_{x_j}(x_k), \frac{P_{x_j} x_i}{|P_{x_j} x_i|} \rangle \\
&\geq -Ce^{\beta(m_t - d_t)} \geq -Ce^{-\beta(1-\cos(\alpha/4))}.
\end{aligned}
$$

- if $i = \underline{i}$ and $j \neq \underline{j}$:

$$\partial_t \arccos(\langle x_{\underline{i}}, x_j \rangle) = - \frac{1}{\sqrt{1 - \langle x_{\underline{i}}, x_j \rangle^2}} \partial_t \langle x_{\underline{i}}(t), x_j(t) \rangle$$

$$= - \frac{1}{\sqrt{1 - \langle x_{\underline{i}}, x_j \rangle^2}} \frac{e^{\beta(1-d_t)}}{Z_\beta(x_{\underline{i}})} \sum_{k=1}^{N} e^{\beta \langle x_{\underline{i}}, x_k \rangle} \langle P_{x_{\underline{i}}}(x_k), x_j \rangle$$

$$- \frac{1}{\sqrt{1 - \langle x_{\underline{i}}, x_j \rangle^2}} \frac{e^{\beta(1-d_t)}}{Z_\beta(x_j)} \sum_{k=1}^{N} e^{\beta \langle x_j, x_k \rangle} \langle P_{x_j}(x_k), x_{\underline{i}} \rangle$$

$$= - \frac{e^{\beta(1-d_t)}}{Z_\beta(x_{\underline{i}})} \sum_{k=1}^{N} e^{\beta \langle x_{\underline{i}}, x_k \rangle} \langle P_{x_{\underline{i}}}(x_k), \frac{P_{x_{\underline{i}}} x_j}{|P_{x_{\underline{i}}} x_j|} \rangle$$

$$- \frac{e^{\beta(1-d_t)}}{Z_\beta(x_j)} \sum_{k=1}^{N} e^{\beta \langle x_j, x_k \rangle} \langle P_{x_j}(x_k), \frac{P_{x_j} x_{\underline{i}}}{|P_{x_j} x_{\underline{i}}|} \rangle$$

$$\geq - \frac{e^\beta}{Z_\beta(x_{\underline{i}})} |P_{x_{\underline{i}}} x_{\underline{j}}|$$

$$- \frac{e^{\beta(1-d_t)}}{Z_\beta(x_{\underline{i}})} \sum_{k \neq \underline{j}}^{N} e^{\beta \langle x_{\underline{i}}, x_k \rangle} \langle P_{x_{\underline{i}}}(x_k), \frac{P_{x_{\underline{i}}} x_j}{|P_{x_{\underline{i}}} x_j|} \rangle$$

$$- \frac{e^{\beta(1-d_t)}}{Z_\beta(x_j)} \sum_{k=1}^{N} e^{\beta \langle x_j, x_k \rangle} \langle P_{x_j}(x_k), \frac{P_{x_j} x_{\underline{i}}}{|P_{x_j} x_{\underline{i}}|} \rangle,$$

by Cauchy-Schwarz inequality, $\langle P_{x_{\underline{i}}} x_{\underline{j}}, \frac{P_{x_{\underline{i}}} x_j}{|P_{x_{\underline{i}}} x_j|} \rangle \leq |P_{x_{\underline{i}}} x_{\underline{j}}|$, hence:

$$\geq - \frac{e^\beta}{Z_\beta(x_{\underline{i}})} |P_{x_{\underline{i}}} x_{\underline{j}}| - \frac{e^\beta}{Z_\beta(x_{\underline{j}})} |P_{x_{\underline{j}}} x_{\underline{i}}|$$

$$- \frac{e^{\beta(1-d_t)}}{Z_\beta(x_{\underline{i}})} \sum_{k \neq \underline{j}}^{N} e^{\beta \langle x_{\underline{i}}, x_k \rangle} \langle P_{x_{\underline{i}}}(x_k), \frac{P_{x_{\underline{i}}} x_j}{|P_{x_{\underline{i}}} x_j|} \rangle$$

$$- \frac{e^{\beta(1-d_t)}}{Z_\beta(x_j)} \sum_{k=1}^{N} e^{\beta \langle x_j, x_k \rangle} \langle P_{x_j}(x_k), \frac{P_{x_j} x_{\underline{i}}}{|P_{x_j} x_{\underline{i}}|} \rangle$$

$$\geq \partial_t \arccos(\langle x_{\underline{i}}, x_{\underline{j}} \rangle) - C e^{\beta(m_t - d_t)}.$$

And in the last line we used that:

$$\partial_t \arccos(\langle x_{\underline{i}}, x_{\underline{j}} \rangle) = - \left( \frac{e^\beta}{Z_\beta(x_{\underline{i}})} |P_{x_{\underline{i}}} x_{\underline{j}}| + \frac{e^\beta}{Z_\beta(x_{\underline{j}})} |P_{x_{\underline{j}}} x_{\underline{i}}| \right) + O(e^{\beta(m_t - d_t)}).$$

In both cases the following holds:

$$\partial_t \arccos(\langle x_i, x_j \rangle) \geq \partial_t \arccos(\langle x_{\underline{i}}, x_{\underline{j}} \rangle) - C e^{-\beta(1-\cos(\alpha/4))},$$

hence:

$$\arccos(m_t) - \arccos(m_0) \geq \arccos(d_t) - \arccos(d_0) - C e^{-\beta(1-\cos(\alpha/4))} T,$$

that implies

$$\arccos(m_t) - \arccos(d_t) \geq \arccos(m_0) - \arccos(d_0) - C e^{-\beta(1-\cos(\alpha/4))} T \geq \frac{\alpha}{2}.$$

We can conclude:

$$d_t - m_t \geq 1 - \cos(\alpha/2) > 1 - \cos(\alpha/4).$$

This is sufficient to close the continuation argument. □

**Remark C.2.** *We used the fact that, if $\arccos(x) - \arccos(y) \geq \alpha$, then $y - x \geq 1 - \cos(\alpha)$.*

Now, recall Proposition 3.11:

**Proposition C.3.** *The solutions $x_i(t)$ of the ODE system (SA), under Assumptions 4 and positive $V$, with the rescaled time $dt = e^{\beta(1 - \langle x_{\underline{i}}, x_{\underline{j}} \rangle)} ds$, converge as $\beta \to \infty$ to the solutions of the system:*

$$
\begin{cases}
\dot{y}_k(t) = \begin{cases}
P_{y_{\underline{i}}}(y_{\underline{j}}) & \text{if } k = \underline{i}, \\
P_{y_{\underline{j}}}(y_{\underline{i}}) & \text{if } k = \underline{j}, \\
0 & \text{otherwise,}
\end{cases} \\
y_i(0) = x_i(0)
\end{cases}
$$

*on finite intervals $[0, T_\epsilon]$, with $T_\epsilon$ such that $\langle y_{\underline{i}}, y_{\underline{j}} \rangle \leq 1 - \epsilon$ throughout the interval, for any $\epsilon > 0$.*

First we need the following lemma:

**Lemma C.4.** *If $\beta$ is large enough then $\delta_t := \langle y_{\underline{i}}, y_{\underline{j}} \rangle \leq 1 - c$ on $[0, T]$ implies $d_t = \langle x_{\underline{i}}, x_{\underline{j}} \rangle \leq 1 - c/2$ on $[0, T]$.*

*Proof.* We proceed again using a continuation argument. The derivatives of the differences are bounded by:

$$
\partial_t(\delta(t) - d(t)) = \partial_t \langle y_{\underline{i}}, y_{\underline{j}} \rangle - \partial_t \langle x_{\underline{i}}, x_{\underline{j}} \rangle
$$

$$
= 2|P_{y_{\underline{i}}} y_{\underline{j}}|^2 - \Big( \frac{e^{\beta(1-d_t)}}{Z_\beta(x_{\underline{i}})} \sum_{k=1}^N e^{\beta \langle x_{\underline{i}}, x_k \rangle} \langle P_{x_{\underline{i}}}(x_k), x_{\underline{j}} \rangle
$$

$$
+ \frac{e^{\beta(1-d_t)}}{Z_\beta(x_{\underline{j}})} \sum_{k=1}^N e^{\beta \langle x_{\underline{j}}, x_k \rangle} \langle P_{x_{\underline{j}}}(x_k), x_{\underline{i}} \rangle \Big)
$$

$$
\leq 2|P_{y_{\underline{i}}} y_{\underline{j}}|^2 - 2 \frac{e^\beta}{e^\beta + e^{\beta d_t}} |P_{x_{\underline{i}}}(x_{\underline{j}})|^2 + Ce^{-\beta(1-\cos(\alpha/4))}
$$

$$
\leq 2|P_{y_{\underline{i}}} y_{\underline{j}}|^2 - 2 \frac{e^\beta}{e^\beta + e^{\beta d_t}} |P_{x_{\underline{i}}}(x_{\underline{j}})|^2 + Ce^{-\beta(1-\cos(\alpha/4))}
$$

$$
= 2|\delta_t(t) - d_t(t)|^2 + 2\Big| \frac{e^\beta}{e^\beta + e^{\beta d_t}} - 1 \Big| + Ce^{-\beta(1-\cos(\alpha/4))}
$$

$$
\leq 2C|\delta_t(t) - d_t(t)| + 2e^{-\beta c/2} + Ce^{-\beta(1-\cos(\alpha/4))}.
$$

where we used $|P_x(y)|^2 = 1 - \langle x, y \rangle^2$ and the previous lemma. The conclusion is again an application of Gronwall's lemma. $\qquad\square$

*Proof of Proposition 3.11.* Thanks to the previous lemma for $\beta$ large enough, on $[0, T_\epsilon]$ we have $d_t < 1 - c_\epsilon$. Now we can proceed with the proof:

Consider the case $k = \underline{i}$.

$$
|x_{\underline{i}}(t) - y_{\underline{i}}(t)| = \int_0^t |\dot{x}_{\underline{i}}(s) - \dot{y}_{\underline{i}}(s)| ds = \int_0^t \Big( \frac{e^{\beta(1-d_s)}}{Z_\beta(x_{\underline{i}})} \sum_{k=1}^N e^{\beta \langle x_{\underline{i}}, x_k \rangle} P_{x_{\underline{i}}}(x_k) - P_{y_{\underline{i}}}(y_{\underline{j}}) \Big) ds
$$

$$
\leq \int_0^t \Big| \frac{e^\beta}{Z_\beta(x_{\underline{i}})} P_{x_{\underline{i}}}(x_{\underline{j}}) - P_{y_{\underline{i}}}(y_{\underline{j}}) \Big| ds + Ce^{-\beta(1-\cos(\alpha/4))}
$$

$$
\leq L \int_0^t |x_{\underline{i}} - y_{\underline{i}}| + |x_{\underline{j}} - y_{\underline{j}}| ds + Ce^{-\beta(1-\cos(\alpha/4))} T.
$$

where we used the previous lemma and the fact that $\frac{e^\beta}{Z_\beta(x_i)} \approx 1 - e^{-\beta(1-d_s)} \approx 1$ thanks to the propertyon $T_\epsilon$. The case $k \neq i, j$ is similar. Hence:

$$\sum_{k=1}^{N} |x_k(t) - y_k(t)| \leq L \int_0^T \sum_{k=1}^{N} |x_k(t) - y_k(t)| + Ce^{-\beta(1-\cos(\alpha/4))}T.$$

The conclusion is then just an application of Gronwall's lemma. $\qquad \square$

## D   Useful lemmas

**Lemma D.1.** *Let $k > 0$, $\beta \to \infty$. Then:*

$$\int_{\mathbb{S}^{d-1}} (1 - \langle x, y \rangle)^{\frac{k}{2}} e^{\beta \langle x, y \rangle} dy \sim C_{d,k} \beta^{-\frac{d-1+k}{2}} e^\beta.$$

*Proof.* In the following $C_{d,k}$ is a constant that depends just on the dimension and on $k$ and could change at each line:

$$\begin{aligned}
\int_{\mathbb{S}^{d-1}} (1 - \langle x, y \rangle)^{\frac{k}{2}} e^{\beta \langle x, y \rangle} dy &= C_{d,k} \int_{-1}^{1} (1 - t)^{\frac{k}{2}} e^{\beta t} (1 - t^2)^{\frac{d-3}{2}} dt \\
&= C_{d,k} \int_0^1 (2 - 2u)^{\frac{k}{2}} e^{\beta(2u-1)} (4u(1-u))^{\frac{d-3}{2}} du \\
&= C_{d,k} e^{-\beta} \int_0^1 (1-u)^{\frac{k+d-3}{2}} u^{\frac{d-3}{2}} e^{2\beta u} du \\
&= C_{d,k} e^{-c} M \left( \frac{d-1}{2}, \frac{k}{2} + d - 1, 2\beta \right),
\end{aligned}$$

where $M$ is the Kummer's confluent hypergeometric function and its asymptotic behavior for $\beta \to \infty$ (see [1]) is given by:

$$I \sim C_{d,k} \beta^{-\frac{d-1+k}{2}} e^\beta.$$

$\qquad \square$

**Lemma D.2.** *Suppose that $T$ is a tensor such that:*

- *$T$ is invariant under permutations of the indices.*
- *$T$ is invariant under rotations that fix a unit vector $x$.*

*Then if $T$ is a 2-tensor, then it must be of the form:*

$$T = \alpha(x \otimes x) + \beta Id.$$

*If $T$ is a 3-tensor, then it must be of the form:*

$$T = \alpha(x \otimes x \otimes x) + \beta Sym(x \otimes I)$$

*Proof.* For simplicity, suppose $d \geq 5$. Without loss of generality we can assume $x = e_1$. Let's start with the case of the 2-tensor. For every $i \neq j$, we can consider another index $l \notin \{1, i, j\}$ and the rotation $R$ such that $Re_i = -e_i$ (wlog $i \neq 1$, otherwise use $j$), $Re_l = -e_l$ and elsewhere is the identity. Then:

$$T[e_i, e_j] = T[Re_i, Re_j] = -T[e_i, e_j],$$

that implies $T[e_i, e_j] = 0$. If $i = j > 1$, then there exists a rotation $R$ such that $Re_i = e_2$, $Re_2 = -e_i$ and the identity elsewhere:

$$T[e_i, e_i] = T[Re_i, Re_i] = T[e_2, e_2]$$

This conclude the proof for the 2-tensor.

For the 3-tensor: consider $i, j, k$ such that $i > 1$ and $(i \notin \{j, k\}$ or $i = j = k)$ . Consider another index $l \notin \{1, i, j, k\}$ and the rotation $R$ such that $Re_i = -e_i$, $Re_l = -e_l$ and identity elsewhere. Then:

$$T[e_i, e_j, e_k] = T[Re_i, Re_j, Re_k] = -T[e_i, e_j, e_k],$$

that implies $T[e_i, e_j, e_k] = 0$. The only cases left are given by $i = j$ and $k = 1$ and their permutations. If $i = j > 1$ and $k = 1$, then construct the rotation R such that $Re_i = e_2$ and $Re_2 = -e_i$ to conclude, as above, that the tensor must be of the form $\alpha(x_i \delta_{jk} + x_j \delta_{ik} + x_k \delta_{ij}) + \beta x_i x_j x_k$. ☐

## E  Supplementary figures

This experiment uses the same settings as in Figure 2 in the backward regime, but with an initial distribution given by a mixture of four wrapped Gaussians on the xy-plane. The red curve shows the interaction energy of the system over time on a logarithmic timescale. We highlight the three distinct timescales and the corresponding behaviors discussed in the paper.

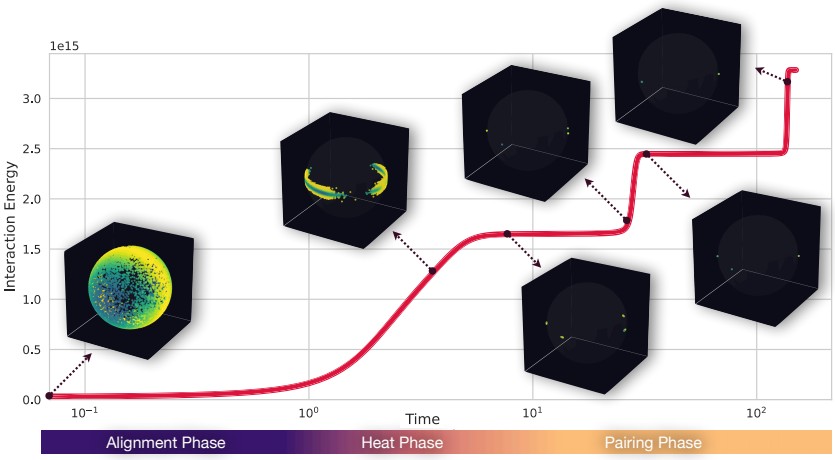

Figure 4: Evolution of the dynamics with $Q^t K = V = S$ definite positive.

