# OpenReview forum: "A multiscale analysis of mean-field transformers in the moderate interaction regime"
_NeurIPS.cc/2025/Conference — NeurIPS 2025 oral_

### Official Review · Reviewer_uk1K · 2025-07-01

**Clarity:** 3
**Significance:** 4
**Originality:** 4
**Rating:** 6
**Confidence:** 4

**Summary:**

This paper presents a theoretical analysis of the token dynamics in a continuous-depth, mean-field transformer model. The authors study the "moderate interaction regime", where the parameters $N, \beta$ tend to infinity together. The central contribution is the identification of a multi-scale dynamical structure of the system, composed of three distinct phases:

1. Alignment Phase ($O(1)$ timescale): A novel finding where the tokens rapidly align with a low-dimensional subspace ($E_{\max}$) determined by the leading eigenspace of the matrix $V K^{\top}Q$. The dynamics are shown to converge to a specific linear transport equation, that is significantly simpler than the original equation. The assumptions in this part are very weak, noticeably so compared to other results in the field. The statement itself is a groundbreaking result for the field.
2. Heat Phase ($O(\beta)$ timescale): On a slower timescale and under more restrictive assumptions, the dynamics evolution within $E_{\max}$ are approximated by a forward or backward heat equation, leading to either smoothing or clustering of tokens.
3. Pairing Phase ($O(e^{c \beta}$ timescale): On an exponentially long timescale, pre-formed clusters are shown to merge sequentially, with the closest pair collapsing along a geodesic.

Authors provide rigorous proofs for stated results, and the statements combined together contextualize the dynamics into a unified and clear, albeit idealized, picture of token evolution. It is supported by some numerical experiments, that are aligned with provided picture.

**Questions:**

**1. Transition from the Alignment to the Heat Phase.**
    The analysis of the Heat Phase assumes that the system's state is fully supported on $E_{\max}$. However, the convergence is asymptotic, suggesting that at finite time, residual parts of the measure may exist outside $E_{\max}$. Could you please elaborate if this impact the dynamics significantly?

**2. Numerical experiments**
    The pictures present justify the complete picture. Could you please check other variations of matrices Q, K, V to see if that is still true? For instance, take $Q^\top K = -Id$ in Figure 1a, or otherwise misaligned $V$ and $Q^\top K$ in Figure 2. I believe the initial phase is still intact, while others are not entered under such assumptions.

 **3. Conditions on $\beta**
    It is hard to track what is assumed of $\beta$ exactly. For example, in referenced SSmax approach $\beta \sim \log n$. Could you share if you expect the alignment phase to still hold for say larger values of $\beta$ (as far as I can see, it is assumed to grow very slowly in the paper)?

**Ethical Concerns:**

["NO or VERY MINOR ethics concerns only"]

**Final Justification:**

Great contribution! The paper rigorously proves a complex and interesting phenomena.

**Limitations:**

Yes, authors addressed the limitations of the paper. Assumptions in their statements are clear and the proofs are well-written.

**Paper Formatting Concerns:**

I have no concerns about formatting.

**Quality:**

3

**Strengths And Weaknesses:**

**Strengths:**
1. The paper's most important result is the rigorous identification and characterization of the initial "alignment phase", supported by a complete proof of the statement under rather weak assumptions (invertible Q, K, V). The statement is interesting both theoretically, as an important approximation that simplifies the ODE analysis, and practically, as a strong evidence that early layers of a transformer might perform dimension reduction of the system to an explicitly given eigenspace. This observation alone gives rise to many questions: to check what said space might be in real transformers, whether it is shared by different layers, and others. In theoretical research, it simplifies the analysis for the system with general parameters. This makes it an important result that deserves recognition to foster further research.
The proof is complex and technically impressive. Because of weak assumptions, it might be a more general phenomenon (that holds under weaker assumptions on N, beta), making it a compelling explanation for rapid dimensionality reduction in transformers.
2. The alignment phase, together with Laplace approximation of the dynamics, is an original observation about the system with important consequences. It is clearly conveyed in the paper and can be clearly tracked, providing clarity of exposition.
3. The overall paper layout provides interesting picture of the tokens geometry evolution on various time scales. Second and third phases are important for overall understanding of the dynamics. Their analysis is not completely novel, as it in many parts coincides with established results, but it provides clarity and completeness to the paper. Overall presentation is well-written and gives a clear picture about the system.
4. The paper is of high quality and well-written. It presents novel and significant results, while being easy to read. Its clarity is supported by good organization and experimental pictures.

**Weaknesses:**
1. The phase transitions between Alignment, Heat, and Pairing phases are idealized. The unified narrative relies on simplifying assumptions at the transition between phases. Most notably, the analysis of the Heat phase assumes the system's state is already fully confined to the subspace $E_{\max}$. This overlooks the potential impact of residual components outside $E_{\max}$ after the alignment phase, which could alter the next slower dynamics.
2. Sequentially presented stages Alignment -> Heat -> Pairing might lead readers to believe that this is a complete description of the dynamics. In particular, this would imply that the final configuration is present in $E_{\max}$. This does not actually follow from the paper results. In addition, presented numerical simulations seem to agree with that conclusion, but they rely on some specific properties of $Q, K, V$. For example, taking $Q^\top K = - Id$ in the Figure 1a experiment (with the same $V$ and other parameters) leads to a similar initial picture (alignment phase) and a different final token position (because alignment does not happen completely and tokens continue to be influenced by components outside $E_{\max}$). In general, importance of $E_{\max}$ is clear for the initial stage, but it does not automatically transfer to later stages.

---

> ### Author Rebuttal · Authors · 2025-07-31
>
> We thank the reviewer for their thorough review and for their positive assessment of our work. We discuss their questions below:
>
> - **[Q1]**: We agree with the reviewer about the importance of rigorously connecting the phases presented in the paper:
> while our analysis suggests that in certain stable regimes the full three-phase picture holds (for example in the backward regime, when $Q^T K = S = V$ for a positive definite matrix $S$, from our simulations and from the existence of an energy function and the fact that Dirac deltas on $E_{\max}$ are stable equilibria [9]), and that the three phases are  achieved sequentially, a rigorous unification of the different dynamical phases for general parameter choices is indeed a challenging open problem, central to which is the stability of the subspace $E_{\max}$.  Due to space constraints, we could only briefly touch upon this in Remark 3.5 and our conclusions, but we now expand in section 5 on the need to identify under which conditions the transition from alignment to heat phase observed in our experiments can be made rigorous, increasing the transparency of the paper.
>     -  Line 334: A further direction of future research consists in providing sufficient conditions for the stability of the space $E_{max}$ emerging in the alignment phase under the prelimit model (i.e., for large but finite $\beta$), thereby justifying assumption 3 and, ultimately, connecting in a rigorous way the alignment and heat phases identified in this paper.
> - **[Q2]**: We thank the reviewer for suggesting an interesting choice of parameters ($Q^TK=-Id$ and same parameters of Figure 1a).
> We have run additional simulations for this case. As the reviewer anticipated, the alignment phase is still achieved, with the tokens collapsing to the predicted y-axis. However, after this initial phase, finite-$\beta$ effects, which depend on the order of the $\beta, t \to \infty$ limits, cause the two clusters to merge along the z-axis. While the example provided by the reviewer does not satisfy Assumption 4, and therefore we make no claims in our paper on such a choice of parameters, we agree that, connecting to the previous question, exclusively based on the results of the paper we cannot rule out situations where the long-term dynamics of the system differs from our results because, e.g., of residual parts of the measure outside $E_{\max}$ and the stability of this subspace.
> We stress again that investigating this phenomenon across different parameter regimes is a crucial and exciting future research direction. We believe this is key to providing a more complete picture of the dynamics, including the unstable cases, and to extend the picture presented in our paper to more general and realistic choices of parameters.
> Other numerical experiments also highlight other types of finite-$\beta$ effects. For instance, in the forward diffusion case, the alignment and heat phases are still visible, but their separation is less sharp. We also note that in the backward case, if the initial measure collapses to a uniform distribution on the $xy$-plane (an unstable equilibrium), the heat phase effectively disappears, and the system proceeds to the clustering behavior described in [7].
>
> - **[Q3]**: The exact maximum scaling rate of $\beta$ with respect to $N$ depends on several factors, including the spectral properties of the matrices $Q, K, V$ and the initial condition. We opted not to provide a specific rate in the paper because the technique used in our proof, particularly in Section 3.1, although simplifying certain mathematical steps, yields to a rate that we believe to be suboptimal. A naive calculation based on our approach gives a very slow growth of order $O(\log(\log(N)))$ and numerical experiments would require an extremely large amount of GPU memory. However, we have strong reasons to believe there is significant room for improvement. For comparison, in the related problem of convergence of a model analogous to the noisy unnormalized self-attention to the porous medium equation in $\mathbb{R}^d$, Oelschläger [32] established convergence for a much faster scaling of $\beta = O(N^{\alpha/d})$ for $\alpha \in (0,1)$. We conjecture that a more refined analysis could establish a similarly improved rate for our setting.
>
> We again thank the reviewer for their positive assessment and their insightful comments. We are happy to continue discussing in the next phase of the reviewing process.

---

> > ### Comment · Reviewer_uk1K · 2025-08-05
> >
> > I thank the authors for answering my questions. I hope they provided valuable feedback to improve clarity of the paper. I will keep my initial score.

---

> > > ### Author Response · Authors · 2025-08-06
> > >
> > > We thank the reviewer for their constructive feedback and in-depth questions. The points raised were helpful in guiding us to clarify several aspects of the discussion of our results. We believe these revisions have meaningfully improved the transparency and rigor of the paper.

---

### Official Review · Reviewer_7zc7 · 2025-07-02

**Clarity:** 4
**Significance:** 4
**Originality:** 4
**Rating:** 6
**Confidence:** 5

**Summary:**

The authors study signal propagation in Transformers, which are now well-accepted as interacting particle systems on the unit sphere (due to layer normalisation). They consider the continuous time perspective, in which time plays the role of layers, and study the appearance of multiple time scales of clustering, in which tokens go from context length equal to n at initial time (layer 0), to several clusters (k<n, say), to a single cluster in infinite time, over multiple time scales. This behavior has already been observed empirically in several previous works, and this paper is one of the first which completely elucidates the transition from n to k points, along with the infinite time collapse to a single point, in a particular scaling of the temperature parameter ß (a magnitude of the key-query matrices) and the context length.

**Questions:**

I have few questions and remarks for the authors which I’d like to see clarified.

1. The pairing phase, in which ß goes to +\infty, is very remeniscent of the results presented in Section 6 of the paper by Geshkovski, Koubbi, Polyanskiy and Rigollet. In fact, the time re-scaling is almost identical, and their result describes the full collision dynamics over all time (albeit, with a slightly modified model). The authors should discuss this more and provide more comparisons.

2. An example after Proposition 3.9 of initial data is provided. Perhaps the authors can comment on the fact that any positive Gevrey-1/2 initial datum also yields the desired assumption (I think), because the equation satisfies a maximum principle. If I am mistaken, I welcome the authors to correct me.

3. In the « heat phase », the authors use the scaling to get a limit to the heat equation, for which attention dynamics are a particle method of. The authors also very nicely comment on the fact that unnonmalized attention dynamics are a particle method of the porous medium equation. Do the results presented in this paper also transfer to unnormalised attention dynamics? I wonder if in the regime in which Q^T K \neq I_d, the partition function is not just a scaling factor, but plays a more important role (some kind of homogenization?)

4. Regarding the presentation:

- Perhaps a simple figure in which the time variable is considered in the horizontal axis, and the different results/phases are superposed, in page 4, would further help the detection of the different phases.

- There are some typos here and there: \R\geq0 in (2), ..

**Ethical Concerns:**

["NO or VERY MINOR ethics concerns only"]

**Final Justification:**

The authors have answered all of my questions appropriately. All issues are resolved.

**Limitations:**

yes

**Quality:**

4

**Strengths And Weaknesses:**

This is one of the best mathematical papers I have seen in NeurIPS over the past 5 to 10 years. The mathematics are deep and different from almost anything used in the machine learning theory literature (deep insights from partial differential equations), and the model is neither trivial nor over-simplified. The results are truly excellent both from an applied and theoretical point of view. I would be surprised if this is not the best theoretical submission of the year.

---

> ### Author Rebuttal · Authors · 2025-07-31
>
> We thank the reviewer for their supportive comments and their appreciation of our work. We address their questions below:
>
> - **[Q1]**: We fully agree with the reviewer that the dynamics of the pairing phase are closely related to the results of Geshkovski, Koubbi, Polyanskiy, and Rigollet [20].
> Indeed, the work in [20] offers a detailed and rigorous treatment of the long-timescale collision dynamics, and we have updated the manuscript with a more thorough discussion of the connection, clarifying the parallels in the analysis and directing interested readers to that reference for an in-depth study of this regime. While [20] was cited in our original submission, we now expand on the previously incomplete discussion comparing our results with those in [20]:
>     - Line 274: "We note that
> this hardmax particle interaction, as well as the timescale where it arises in the large $\beta$ limit, was
> introduced in [20, Section 6] in the case $d = 2$. We present an analogous result here in arbitrary
> dimension, without claiming originality, to provide a complete dynamical picture across phases."
>     - Line 279: "Note that this result only holds up to an
> arbitrary moment before the first collapse. We refer to [20, Section 6] for a detailed explanation of
> the challenges to bypass this limit and a proof of an analogous result until and beyond the collapse time in a related but simplified model. The above proposition is proven for completeness in Appendix C."
>
> - **[Q2]**: The reviewer’s observation is indeed correct. Positive initial data in the Gevrey-1/2 class satisfies the required assumptions for both the forward and backward problems: globally-in-time in the forward case, due to the maximum principle, and locally-in-time in the backward direction, thanks to Hadamard-type stability results in the Gevrey-1/2 class (applicable to the backward heat equation).
> We will explicitly mention this fact in the revised version:
>     - Line 250: “A more general class in which local existence and well-posedness hold in both the forward and backward directions is the set of positive, Gevrey-1/2 functions.”
>
> - **[Q3]**: While a similar multi-scale approach could be applied to the unnormalized self-attention model, leading to a different set of limiting dynamics as the reviewer rightfully predicts, we expect that this extension would entail a slightly different set of mathematical challenges:
>     - The alignment phase would no longer be described by a linear transport PDE. Instead, the limit would formally be a non-linear transport equation, roughly $\partial_t \mu = - \text{div}(\mu V[\mu])$, where the dependence on $\mu$ of the vector field $V\[\mu\](x)=\mu(\frac{K^TQx}{|K^TQx|})P_xV(\frac{K^TQx}{|K^TQx|})$ arises from the unnormalized attention. Establishing a collapse result (similar to Proposition 3.4) for this equation has proven to be more challenging than in the normalized model because of the non-local structure of the limiting PDE.
>     - The heat phase would lead to a porous medium equation (PME) with exponent 2. Interestingly, proving convergence to the PME is more straightforward than in our case, thanks to the Wasserstein gradient flow structure of the USA model proved in  Geshkovski, Letrouit, Polyanskiy, and Rigollet [21].
>
> From this comparison, one could conclude that in this regime, the normalization factor in self-attention acts as a "linearization" of the flow from a PDE perspective. It transforms the non-linear transport equation into a linear one and the PME into the heat equation. Because of this,  we indeed expect a more complex interaction structure to arise in the limiting model in the unnormalized case.
>
> - **[Q4]**:  We thank the reviewer for these helpful suggestions. A timeline figure illustrating the different dynamical phases would greatly improve the paper's readability.
> We have plotted the evolution of the interaction energy $\sum_{i,j} \exp(\beta \langle x_i, x_j\rangle)$ for an experiment with initial condition given by a superposition of four gaussian distributions and $Q^tK=V=diag(1,1,0.1)$, corresponding to the backward regime, as a function of time on a logarithmic scale, where the three phases identified in the paper appear as different rates of increase of such energy.  We also appreciate the reviewer for pointing out the typos. We will correct them throughout the manuscript.
>
> We stand by for further discussion on the interesting questions asked by the reviewer in the next phase of the reviewing process.

---

> > ### Comment · Reviewer_7zc7 · 2025-08-05
> >
> > I thank the authors for the very detailed answers to all of my questions. I have no further comments and my initial opinion remains intact.

---

> ### Author Response · Authors · 2025-08-05
>
> We are grateful to the reviewer for the encouraging comments and insightful questions. We’re glad that the main contributions and motivations of our work were well received, and we greatly appreciate their endorsement.

---

### Official Review · Reviewer_dHkY · 2025-07-03

**Clarity:** 2
**Significance:** 2
**Originality:** 3
**Rating:** 5
**Confidence:** 3

**Summary:**

This paper analyzes the evolution of transformers inference time representations in in the mean field interaction regime, where the number of tokens ($N$) and softmax temperature ($\beta$) scale together. The authors identify three distinct phases that emerge depending on the order of magnitude of the chosen timescales, as a function of $\beta$, with the characterization of the "alignment phase" on a fast timescale being the main novelty. Additionally, the work serves to combine it, under simpler assumptions, with the intermediate "heat phase" and an exponentially long "pairing phase," into a single dynamical picture. Finally, the work is supported by numerical simulations.

**Questions:**

- With respect to the heat phase, I understand that $\gamma$ controls the forward vs backward diffusion. The analysis of the Pairing Phase proceeds under the assumption of a positive V, which corresponds to the clustering (backward diffusion) regime. Could the authors comment on the long-term behavior in the forward diffusion regime ($\gamma$<0)? How does one determine which regime the model evolves through in the heat phase (in other words, why is gamma assumed to be positive)?
- Akin to [1], is possible to have simulations with actual models to better justify the proposed system and its assumptions?


References
- [1] The emergence of clusters in self-attention dynamics. Geshkovski et al., 2024.

**Ethical Concerns:**

["NO or VERY MINOR ethics concerns only"]

**Final Justification:**

Following discussion with the authors and their efforts towards addressing my concerns about the presentation, as well as reading through other reviews and the discussion, I am confident in recommending this paper for acceptance, increasing my score 4->5

**Quality:**

3

**Strengths And Weaknesses:**

I will preface my review by saying that I am likely not the intended audience for this work, due to limited familiarity with related literature or proof techniques. Therefore, I structure my review largely from the point of view of relevance and accessibility for the general audience.

Strengths:
- The proposed contributions of the paper are listed clearly and appear novel, and the paper seems well placed within the context of related work.
- The analysis appears comprehensive and presents interesting insights into the evolution of transformer representations

Weaknesses:
- My biggest qualm is that this paper often assumes a very high level of notational and technical familiarity, and pursuit of rigor sometimes leads to sacrificing the high level perspective. I list a few instances below:
    - It is unclear what the inverse sampling temperature (which generally only relates to the logit layer softmax in popular context) or timescales are denoting in the introduction, until one reads through section 2.
    - While the introduction to section 3 does a decent job of delineating the phases at a high level, there is no attempt to tie them back to any notion of practically observed behaviors in any of the exposition in section 3. It would significantly strengthen the paper if each subsection briefly motivated the analysis by connecting it to a hypothetical but grounded representation learning behavior/goal.
    - For the latter criticism, the exposition around the technical result often focuses on technicalities and does not offer enough high level intuition on the result itself or why it would be important. For instance, the proof sketch of Theorem 3.2 immediately references concepts like the 'Ascoli-Arzelà theorem' and 'cumulants of the Von Mises-Fisher distribution'. While technically precise, this gives the non-specialist reader little intuition about why the system should behave this way. (also see remarks proceeding proposition 3.4, or discourse around proposition 3.9).
- I understand that the model/assumptions would be considered general (or even weaker) in the context of related work. But from a general standpoint, the model still appear fairly unrealistic and it would be helpful to briefly discuss what is lost by these simplifications. For instance, what complexities are lost in Remark 2.1 or Assumption 4?

Overall: Despite my critiques on the exposition, I want to stress that I'm positive about the work's technical contributions. My main suggestion is to make the paper more self-contained to better reach the general conference audience. For example, along with the suggestions above, perhaps adding some background on key concepts in the appendix could significantly improve accessibility and let non-familiar readers appreciate the novelty of this work. My score is based on these presentational concerns, and I am willing to champion this paper if these are addressed in the rebuttal.

---

> ### Author Rebuttal · Authors · 2025-07-31
>
> We thank the reviewer for the valuable suggestions aimed at improving the presentation of our work to make it more accessible to the general audience of the conference.
> - **Weaknesses**: To address their punctual remarks, we applied the following modifications to an updated version of the paper:
>     - Line 18: "...choosing the model’s inverse temperature $\beta$ (a parameter that scales the query-key dot products in the self-attention layer) as a function of N [31, 36]."
>     - Line 168: "Under these conditions, we show that the limiting dynamics in this regime coincides with the formal Laplace approximation of term (I) in (4), i.e., the integrals in the definition of the vector fields can be replaced by the value of the integrand at the maximum point $x' = K^T Q x/|K^T Q x|$, leading to the significantly simplified expression (5) below."
>     - Line 171: "To establish the result we must prove well-posedness of the family of equations leading to the desired limit and obtain sufficient regularity uniformly in $\beta$ to ensure that the formal simplifications from (4) (I) to (5) are allowed. This is particularly important as the derivatives of the kernel tend to infinity in the limit $\beta \to \infty$."
>     - Related works: we added references to the literature on rank collapse and oversmoothing to connect the geometric phenomena described in the paper to more applied consequences.
>     - Remark 2.1: the MLP would act as a drift term in the dynamics, whose consequence should still be investigated further. We expect different dynamical behavior depending on the relative scale of the MLP coefficients and the attention part.
>     - Conclusions and limitations: we have expanded in the paper both on the potential practical implications and limitations of the paper, summarizing the points in the answer to Reviewer aiFK.
>     - Appendix: we included paragraphs preceding each technical result and lemma providing a high level explanation of the result, its role in our proof and referring the reader, when necessary, to complementary external literature on the topic.
>     - Regarding the timescales, they can be viewed as a conceptual framework illustrating how transformers progressively build understanding.  From this perspective, the process proceeds in three distinct parts: an initial, fast Alignment Phase establishes a global context by collapsing tokens into a shared feature space. This is followed by an intermediate Heat Phase, where these representations are refined into discrete semantic clusters. The final, slow Pairing Phase then constructs a conceptual hierarchy by merging these clusters to form higher-order abstractions.
> While we believe this interpretation is a suggestive lens through which to view our results, we are cautious about including such speculation in the main body of the paper. Our primary focus is the rigorous, provable mathematical treatment of the model's dynamics, and we wish to focus on rigorous contributions rather than providing heuristics that can't be substantiated by mathematical proofs.
>
> We are happy to further exchange with the reviewer during the discussion phase if they believe there is further space for improvement.
>
> - **[Q1]**: In the forward regime, the heat phase proceeds in the opposite direction, spreading the particles far apart and reaching the unique steady state of the forward heat equation, which is the uniform measure on $E_{max}$. This is an equilibrium of the model, so there is no pairing phase in this case, and our theoretical analysis concludes there (justifying our assumption for $\gamma > 0$ in the pairing phase). What one could expect in numerical simulations with a large but finite $\beta$ is that the particles spread around $E_{max}$, and then because of finite-$\beta$ effects or the possible instability of $E_{max}$ beyond this timescale, new phenomena that are not described by our analysis may occur (see also answer to Q1, Q2 of reviewer uk1K).
> - **[Q2]**: In [22] ([1] in the review), the authors provide numerical simulations with ALBERT.  While we stress that our model is designed to isolate the core self-attention mechanism and, as the reviewer rightly points out, does not yet incorporate other architectural features of models like ALBERT,
>  we can still draw a parallel between the phenomena in ALBERT and our description of the dynamics by running a simulation where we increase the inverse temperature of the model tested in [22] to bring it in the regime studied in our paper. Since the clustering tendency is studied in [21], our simulation aims at establishing the presence of the alignment phase in the above model.
> While we cannot upload the results of the simulation because of this year's review policy, they show a rapid collapse of the effective dimension of the tokens in the first layers of the model. More specifically, we added to the appendix a graph depicting a sudden collapse of the effective dimension of the token empirical measure as a function of layer depth in the first layers of the model (the alignment phase) followed by a slower decrease in the heat phase.

---

> > ### Author Response · Authors · 2025-08-03
> >
> > Adding to our previous responses and to further encourage discussion in case our reply did not sufficiently address the reviewer’s remarks, we would like to clarify some additional points related to our rebuttal by including the following changes in the manuscript:
> > - Line 180: A consequence of Theorem 3.2 is that in the large-$\beta$ limit, the tokens, to leading order, evolve independently of each other, driven primarily by the structure of the $Q$, $K$, and $V$ matrices. In this regime, self-attention behaves like a composition of linear layers followed by layer normalization, with minimal influence from inter-token interactions.
> > - Line 193: Proposition 3.4 demonstrates that the token representations rapidly collapse onto a lower-dimensional subspace determined by the model's matrices. This can be interpreted as the initial phase of the inference process, where information is compressed into a smaller, more relevant subspace.
> > - Line 261: The dynamics in this phase, governed by a heat equation, drive the formation of distinct token clusters (via backward diffusion) or the smoothing of the token distribution (via forward diffusion). This behavior can be interpreted as a representation refinement stage, where tokens are organized into more defined semantic groups.
> > - Line 279: This final, slow phase models the sequential merging of the closest token clusters. This can be interpreted as the construction of higher-order abstractions, where previously formed groups are hierarchically combined to create more complex representations.
> > - Conclusions (Remark 2.1 and Assumption 4): There are several important directions in which our work could be extended. Most notably, incorporating the MLP, which could be interpreted as introducing a drift term in the dynamics, acting independently on each token without accounting for mutual interactions. Another natural extension involves studying the dynamics under more general parameter settings. For instance, during the heat phase, we assume that $Q^t K = S$ is symmetric positive definite, which holds, for example, when $Q = K$. This “shared-QK” assumption is not novel and has been adopted in prior empirical work (e.g., Kitaev et al., Reformer: The Efficient Transformer). While different choices of these parameters (both MLP and Attention matrices) can have a dramatic effect on model behavior, with adversarial choices potentially leading to qualitatively different dynamics from the one predicted in this paper, we believe our results to be a relevant first step towards understanding the development of representations in transformers, capturing some important qualitative features of these models as shown in [21].
> >
> > We hope that our additional comment, based on what we believe is a more accurate interpretation of the reviewer's remarks, satisfactorily addresses some of the points they raised. We would warmly welcome their feedback on this matter in case our interpretation is not yet fully in line with their comments.

---

> > > ### Comment · Reviewer_dHkY · 2025-08-03
> > > **Response to Rebuttal & Comment**
> > >
> > > I am satisfied by the author's efforts to improve the clarity of the presentation. I believe these changes will significantly benefit the general audience, and I am happy to increase my rating for the paper accordingly.

---

> > > > ### Author Response · Authors · 2025-08-05
> > > >
> > > > We would like to sincerely thank the reviewer for their constructive feedback on the presentation and for the updated score. We appreciate their thorough review and their comments, which we feel have improved the quality of the paper.

---

### Official Review · Reviewer_aiFk · 2025-07-06

**Clarity:** 2
**Significance:** 2
**Originality:** 2
**Rating:** 4
**Confidence:** 2

**Summary:**

This paper gives a mean-field analysis of encoder-only transformer models -- it shows that if the number of tokens $N$ is large, and the inverse temperature parameter $\beta$ scales with $N$ slowly enough, the dynamics display a multi-scale behavior -- the token empirical measure collapses onto a low-dimensional subspace at first in the "fast" phase, then collapses into clusters during an intermediate phase, and finally collapses sequentially into a single cluster in the final "slow" phase. The authors rigorously show the limiting dynamics in each of these three phases, and validate them empirically.

**Questions:**

- What does this work tell us about the behavior of modern language models? Is there any immediate practical application?
- Are there potential applications that may arise from further developing this area?
- Why should the LLM community pay attention to mean-field analyses more broadly?
- What are the technical challenges involved in extending the analysis to handle differing Q, K, V matrices in different layers?

**Ethical Concerns:**

["NO or VERY MINOR ethics concerns only"]

**Limitations:**

yes

**Quality:**

2

**Strengths And Weaknesses:**

Strengths:
- Easy to follow paper with a clear set of theoretical and empirical results
- Gaining any theoretical understanding of transformers/deep learning has historically proved challenging, so it's impressive that such an analysis sheds light on some limiting cases
- Compared to previous works which either fail to analyze the intermediate phase characterized by a partial clustering, or do analyze but only in the case where the Q, K and V matrices are multiples of identity, this work analyzes this phase for more general Q, K, V matrices.
- It also handles clustering for hardmax transformers

Weaknesses:
- The analysis seems restricted to the case when all the Q, K, V matrices are the same for the different layers. Furthermore, there are additional assumptions on these matrices. It is not clear how reasonable these assumptions are
- Overall, it's unclear what this analysis really tells us about modern language models and their behavior. No explanation/motivation is provided
- The empirical results again don't seem to really shed any light on modern language models

Overall, while I am skeptical of the practical applicability of the results, it does build upon previous works in the area in a meaningful way, by extending the analysis of the intermediate regime for general Q, K, V matrices. I am not an expert in the area of mean-field analysis.

---

> ### Author Rebuttal · Authors · 2025-07-31
>
> - **[Q1, Q2, Q3]**: We thank the reviewer for the feedback and the opportunity to discuss the potential implications of our work.
>     We agree with the reviewer that some aspects of the original architecture have been simplified in our analysis, given its technically challenging nature, and that, as a consequence, it is not clear that our results apply directly to real world architectures. Still, since a complete theoretical analysis of transformer models remains an open problem, we see our contribution as a step in this direction, that we aim to improve in future works considering increasingly realistic assumptions.
>     While the path from this  theoretical analysis to direct application is not immediate, we recall that analogous
>     clustering behavior in deep transformer networks have been shown to emerge in real-world models [1, 2], and we  believe our findings open several possibilities for future investigation:
>     - **Model interpretability**: our analysis of the alignment phase presents a concrete mathematical mechanism for how token representations might evolve through the depth of transformer networks, predicting their collapse onto subspaces determined by the spectral properties of the $Q, K, V$ matrices, along the same line as what was pointed out by previous works [1, 2, 3]. We believe this finding could open an interesting avenue for model interpretability, allowing researchers to explore the semantic meaning of these learned subspaces.
>     - **A link to diffusion models**: the emergence of the backward heat equation in our analysis reveals an interesting mathematical link between the dynamics of transformers and denoising diffusion models. While the practical implications require further investigation, this connection bridges two of the most important architectures in modern machine learning and may inspire new theoretical insights or hybrid model designs.
>     - **Parameter scaling**: while carried out in an idealized regime, our analysis clearly identifies, from a mathematical point of view, the effect of choosing model hyperparameters at a certain scale on the representations developed by transformer architectures in the given regime. This information, combined and contrasted with analogous studies considering complementary scaling regimes, can inform practitioners on the effect of various choices of such hyperparameters (e.g., scale of inverse temperature, depth, residual connections) on model representations they aim to observe/avoid (analogously to the effect of various choices of parameter scales at initialization for wide feedforward neural networks, resulting in either the NTK or feature learning regimes during the training process [4, 5, 6, 7, 8]).
>
> - **[Q4]**: The main difficulties in extending our analysis to handle matrices that vary from layer to layer are analogous to those encountered when establishing the continuum limit of ResNets as Neural ODEs (as discussed, for instance, in [9]).
> Assuming the matrices $Q_t, K_t, V_t$ vary sufficiently smoothly with depth $t$, one could expect that the alignment dynamics would still be described by a linear transport equation, but with a time-dependent velocity field:
> $$
> \partial_t \mu = - \mathrm{div}\left(\mu P_x \frac{V_t K^T_t Q_t x}{|K_t^TQ_t x|}\right).
> $$
> On the other hand, when the matrices vary rapidly with depth, as is the case at i.i.d. initialization, in the infinite-depth limit considered in our paper we expect an averaging phenomenon to emerge, resulting in dynamics that can be approximated by a system with constant $Q$, $K$, and $V$ matrices.
> However, making these arguments rigorous would require a careful analysis to establish the uniform bounds on the discrete dynamics that are necessary for convergence to this continuous-time limit and is left for future research.
> In any case, we wish to emphasize that the use of shared parameters across layers is a design choice adopted in several models, most notably in [10].
>
>
> Overall, we have improved the conclusion by better highlighting both the limitations and the implications of our work, based on the points discussed above:
> - Line 334: "While the path from this theoretical analysis to direct application is not immediate, we believe our work opens several potential avenues for future investigation. The characterization of the alignment phase, for instance, offers a potential mechanism for interpreting how token representations evolve into learned subspaces. Furthermore, the emergence of the backward heat equation suggests an interesting mathematical link to diffusion models, which may bridge two of the most important architectures in modern machine learning. Finally, by focusing on the "moderate interaction regime", we hope our analysis provides a theoretical foundation for a more principled understanding of parameter scaling, particularly as models are adapted for longer contexts."
>
> We thank again the reviewer for their comments and are happy to continue discussing during the next phase of the review process.
>
> ## Bibliography
>
> [1] Borjan Geshkovski, Cyril Letrouit, Yury Polyanskiy, and Philippe Rigollet. The emergence of
> clusters in self-attention dynamics.
>
> [2] Borjan Geshkovski, Cyril Letrouit, Yury Polyanskiy, and Philippe Rigollet. A mathematical
> perspective on transformers.
>
> [3] Martin Burger, Samira Kabri, Yury Korolev, Tim Roith, and Lukas Weigand. Analysis of
> mean-field models arising from self-attention dynamics in transformer architectures with layer
> normalization.
>
> [4] Raphaël Berthier, Andrea Montanari, and Kangjie Zhou. Learning time-scales in two-layers
> neural networks.
>
> [5] Lenaic Chizat and Francis Bach. On the global convergence of gradient descent for over-
> parameterized models using optimal transport.
>
> [6] Song Mei, Andrea Montanari, and Phan-Minh Nguyen. A mean field view of the landscape of
> two-layer neural networks.
>
> [7] Grant Rotskoff and Eric Vanden-Eijnden. Trainability and accuracy of artificial neural networks:
> An interacting particle system approach.
>
> [8] Jacot, Arthur, Franck Gabriel, and Clément Hongler. Neural tangent kernel: Convergence and generalization in neural networks.
>
> [9] Sander, Michael, Pierre Ablin, and Gabriel Peyré. Do residual neural networks discretize neural ordinary differential equations?.
>
> [10] Lan, Zhenzhong, et al. Albert: A lite bert for self-supervised learning of language representations.

---

> > ### Author Response · Authors · 2025-08-05
> >
> > Adding to our previous responses and to further encourage discussion in case our reply did not sufficiently address the reviewer’s remarks, we would like to refer to the additions to the paper contained in our discussion comment to reviewer dHkY, which may address more in detail also their W1 and W2.
> >
> > In case some of the reviewer’s comments were not satisfactorily addressed in our revision, we would be happy to continue discussing, in the remaining days of the discussion session, how it can be improved further.

---

> > > ### Comment · Reviewer_aiFk · 2025-08-07
> > >
> > > Thank you for your response -- I am still a bit skeptical that the infinite depth regime is the right one to look at from a practical perspective, considering LLMs are not that deep -- I do recognize the difficulty of analyzing real LLMs. That being said, I worry that these kinds of works give the impression that these architectures can be analyzed in meaningful ways to outsiders, when this may not be the case.
> > >
> > > Nevertheless, I appreciate your response, and clarification of what you think may be potential downstream consequences.

---

> > > > ### Author Response · Authors · 2025-08-08
> > > >
> > > > We thank the reviewer for the follow-up. As they note, our work is primarily theoretical and conducted in an idealized regime. This approach is common in the literature, for instance, in Neural ODEs and the Neural Tangent Kernel, where simplified limits have provided valuable insights for the broader community.
> > > >
> > > > To briefly discuss the topic of depth, adding to our previous exchange, we also emphasize that, although our results are derived in the continuous-depth limit, finite-depth Transformers can be interpreted as Euler discretizations of a continuous-time system. This perspective provides theoretical grounds to expect that real models display qualitatively similar behaviors to those predicted by our scaling-limit analysis, which is consistent with the trends observed in the additional experiments we performed in response to the reviewer’s suggestion. Additionally, we note that key qualitative effects (e.g., the experiments on the alignment phase in Fig. 1) can be reproduced with fewer than 50 timesteps (i.e. attention layers), well within the range of state-of-the-art LLMs (for context: DeepSeek-V3 has 61 layers, Qwen3-32B has 64 layers, and Qwen3-235B-A22B has 94 layers).
> > > >
> > > > We therefore believe that such idealized regimes can yield practically relevant insights, and we are looking forward to provide further evidence of this fact in future works. We take this opportunity to thank again the reviewer for their feedback on our paper and remain available for further discussion.

---

### Decision · Program_Chairs · 2025-09-17

**Decision:**

Accept (oral)

**Comment:**

My reading of the paper confirms the reviewers' high scores. This is an excellent paper on a topic of theory of transformers, where contributions are rarely clean and simple to understand. This one discovers a very simple and clean idea of deriving a high-beta versions of the Transformers PDE, which in many ways are more tractable for analysis than the original PDEs of Geshkovski et al. I thus recommend this paper for acceptance and congratulate the authors on this excellent result.